# RSC and GRFs confer promoter directionality by restricting divergent noncoding transcription

Andrew CK Wu[1],*, Claudia Vivori[1],*, Harshil Patel[2], Theodora Sideri[1], Fabien Moretto[1,3], Folkert J van Werven[1]

The directionality of gene promoters—the ratio of protein-coding over divergent noncoding transcription—is highly variable. How promoter directionality is controlled remains poorly understood. Here, we show that the chromatin remodelling complex RSC and general regulatory factors (GRFs) dictate promoter directionality by attenuating divergent transcription relative to protein-coding transcription. At gene promoters that are highly directional, depletion of RSC leads to a relative increase in divergent non-coding transcription and thus to a decrease in promoter directionality. We find that RSC has a modest effect on nucleosome positioning upstream in promoters at the sites of divergent transcription. These promoters are also enriched for the binding of GRFs such as Reb1 and Abf1. Ectopic targeting of divergent transcription initiation sites with GRFs or the dCas9 DNA-binding protein suppresses divergent transcription. Our data suggest that RSC and GRFs play a pervasive role in limiting divergent transcription relative to coding direction transcription. We propose that any DNA-binding factor, when stably associated with cryptic transcription start sites, forms a barrier which represses divergent transcription, thereby promoting promoter directionality.

## Introduction

Transcription is highly pervasive and results in intergenic and intragenic noncoding transcription events. Noncoding transcription and the produced noncoding RNAs play diverse roles in gene and genome regulation (Ard et al, 2017; Gil & Ulitsky, 2020). Transcriptionally active, protein-coding gene promoters are a major source of noncoding transcription. At these genomic locations, noncoding transcription initiates in the divergent direction, a process known as divergent or bidirectional transcription, which generates upstream transcripts from a distinct core promoter in the opposing direction to the coding gene (Seila et al, 2008; Neil et al, 2009; Xu et al, 2009; Sigova et al, 2013). Divergent noncoding transcription is an intrinsic property of coding gene transcription, present across eukaryotes, and widespread across actively transcribed regions in the genome (Seila et al, 2009). Within promoters, the divergent and coding core promoters are pairs of core promoters which typically share the same nucleosome depleted region (NDR) where transcription initiates in divergent directions (Rhee & Pugh, 2012; Scruggs et al, 2015). Moreover, the ratio of divergent over coding transcription gives insights on the directionality of promoters (Jin et al, 2017).

Our understanding of the function of divergent transcription is still incomplete. There is evidence that the noncoding transcripts emanating from divergent transcription events can help to promote gene transcription in the coding direction and to facilitate cell fate control (Luo et al, 2016; Frank et al, 2019; Yang et al, 2021). Other studies have suggested that divergent transcription may constitute transcriptional noise of active promoters (Struhl, 2007; Seila et al, 2009). Promoters, upon fortuitous evolution, are thought to exist in a ground state of relatively balanced divergent transcription. It has been proposed that subsequent mutations arising through evolution restricts transcription in one direction such that promoters become increasingly directional (Jin et al, 2017). As such, evolved promoters are thought to be highly directional and display little divergent transcription but high levels of coding direction gene transcription.

Aberrant noncoding transcription, including divergent transcription, can negatively impact cell fitness. For example, induced noncoding transcription can cause R-loop formation and DNA damage (Nojima et al, 2018). In addition, aberrant divergent transcription events can affect coding gene transcription leading to mis-regulation of gene expression (Chiu et al, 2018; du Mee et al, 2018). This especially impacts species with gene-dense genomes such as *Saccharomyces cerevisiae*. Hence, there are various molecular mechanisms that limit the accumulation of noncoding RNAs or noncoding transcription itself. Indeed, to reduce the accumulation of divergent RNAs, these noncoding transcripts are often rapidly degraded via the RNA degradation machinery (Neil et al, 2009; Xu et al, 2009; Flynn et al, 2011; van Dijk et al, 2011; Malabat et al, 2015). Divergent transcription units are also typically short, because of enrichment in transcription termination signals (Schulz et al, 2013). In addition, divergent transcription can be repressed by

[1]Cell Fate and Gene Regulation Laboratory, The Francis Crick Institute, London, UK    [2]Bioinformatics and Biostatistics, The Francis Crick Institute, London, UK    [3]Institute of Molecular Biology and Biotechnology (IMBB), Foundation for Research and Technology–Hellas (FORTH), Heraklion, Greece

Correspondence: folkert.vanwerven@crick.ac.uk
*Andrew CK Wu and Claudia Vivori contributed equally to this work.

chromatin remodellers and histone-modifying enzymes (Marquardt et al, 2014; Gowthaman et al, 2021 *Preprint*). Lastly, general regulatory factors (GRFs) can repress initiation of aberrant noncoding transcription events (Challal et al, 2018; Wu et al, 2018).

In previous work, we showed that the GRF and pioneer transcription factor Rap1 represses divergent transcription in the promoters of highly expressed ribosomal protein and metabolic genes (Challal et al, 2018; Wu et al, 2018). At these highly directional promoters, Rap1 is crucial for both promoting transcription in the coding direction and for limiting transcription in the divergent direction (Challal et al, 2018; Wu et al, 2018). We proposed that Rap1 limits divergent transcription by interfering with recruitment of the transcription machinery. In the same study, we also identified a role for RSC, a major chromatin remodelling complex important for chromatin organisation (Cairns et al, 1996). We showed for two gene promoters that RSC promotes divergent transcription in Rap1-depleted cells. RSC activity increases nucleosome positioning, which, in turn, is important for maintaining NDRs in promoters (Lorch et al, 1998; Badis et al, 2008; Parnell et al, 2008; Hartley & Madhani, 2009). RSC also acts with GRFs to stimulate transcription in the protein-coding direction (Kubik et al, 2017, 2018; Brahma & Henikoff, 2019). Specifically, RSC positions the +1 nucleosome in gene promoters to allow pre-initiation complex (PIC) assembly and transcription start site (TSS) scanning (Klein-Brill et al, 2019). RSC not only limits aberrant transcription from upstream in promoters in the sense direction but also limits transcription initiating from the 3′ end of gene bodies in the antisense direction (Alcid & Tsukiyama, 2014; Klein-Brill et al, 2019; Kubik et al, 2019; Cucinotta et al, 2021). The role of RSC in controlling promoter directionality remains an area of interest.

Here, we examined the role of RSC and GRFs in controlling promoter directionality. Our analysis reveals that RSC depletion leads to a relative increase in divergent noncoding transcription. The affected promoters tend to be highly directional and are enriched for binding sites of GRFs such as Abf1 and Reb1. Consistent with the role of RSC in chromatin organisation, its depletion affects nucleosome positioning in upstream regions of directional promoters. Finally, we demonstrate that ectopic targeting of GRFs or dCas9 to divergent core promoters can also repress divergent noncoding transcription. We propose that nucleosomes positioned by RSC and GRFs constitute physical barriers which limit aberrant divergent transcription in promoters, thereby increasing promoter directionality.

## Results

### RSC represses divergent transcription independently of Rap1

To investigate how RSC and GRFs promote directionality of transcription more closely, we performed RNA-seq on nascently transcribed RNA (nascent RNA-seq) and on polyadenylated RNAs (mRNA-seq) after depletion of both RSC and Rap1. We determined the levels of nascently transcribed RNA by measuring RNA polymerase II (Pol II)-associated transcripts using an adapted native elongating transcript sequencing protocol (Churchman & Weissman, 2011). In short, we affinity-purified RNA Pol II using Rpb3-FLAG and quantified the Pol II–associated RNAs in wild-type (WT) cells, before and after depletion of RSC and/or Rap1 (Fig S1A). To deplete RSC and Rap1, we used auxin-inducible degron alleles (*STH1-AID* and *RAP1-AID*) and treated the cells with indole-3-acetic acid (IAA, 0.5 mM) for 2 h (Nishimura et al, 2009). Rap1 and Sth1 were efficiently depleted in cells harbouring either one or both degron alleles after treatment (Fig 1A, top panel). For mRNA-seq, we added a spike-in control of *Schizosaccharomyces pombe* cells in a defined ratio. We used the *S. pombe* mRNA-seq signal to normalise mRNA-seq data for *S. cerevisiae* (Fig 1A, bottom panel). The *spike-in* was added to compensate for the effect that RSC depletion has on global transcription by RNA Pol II (Parnell et al, 2008; Klein-Brill et al, 2019). The reproducibility of nascent RNA-seq and mRNA-seq biological repeats for the Sth1 and Rap1 depletions and corresponding controls is shown by the principal component analysis (PCA) in Fig S1B.

Firstly, we examined how a subset of regulated protein-coding transcripts was affected by Rap1 and RSC depletion. We focused on the set of 141 Rap1-regulated genes described previously (Wu et al, 2018) and quantified their expression. As expected, almost all Rap1-regulated genes decreased in expression after Rap1 depletion (*RAP1-AID* + IAA versus *RAP1-AID* + DMSO) (Figs 1B and S1C). Expression of Rap1-regulated genes also decreased in Sth1-depleted cells as detected by mRNA-seq (Fig 1B). We observed no decrease in signal in the nascent RNA-seq for Rap1-regulated genes, likely because internal normalisation was used for the analysis. Because of this, the nascent RNA-seq is to be taken as a relative measurement of transcription levels throughout this manuscript.

Secondly, we examined two previously well-characterised promoters (*RPL43B* and *RPL40B*) at which Rap1 is known to repress divergent transcription (Figs 1C and S1D) (Wu et al, 2018). As expected, transcription of divergent noncoding transcripts (*IRT2* and *iMLP1*) was up-regulated upon Rap1 depletion, whereas coding transcription was decreased. Upon co-depletion of Rap1 and Sth1, we observed that *IRT2* and *iMLP1* levels were reduced in mRNA-seq. Although Sth1 depletion by itself did not affect *IRT2* levels, levels of *iMLP1* were already up-regulated in Sth1-depleted cells (Fig S1D). This suggests that RSC represses divergent transcription relative to coding transcription at this locus. Also, the *RPS10A* gene showed a relative increase in divergent transcription in both Rap1- and Sth1-depleted cells (Fig 1D). Interestingly, divergent transcription in Sth1-depleted cells initiated further upstream from the coding gene compared with Rap1-depleted cells, suggesting that Rap1 and RSC limit divergent transcription through different mechanisms (Fig 1D).

Next, we examined more broadly the promoters of the 141 Rap1-regulated genes for relative changes in divergent transcription (Fig 1E). As expected, Rap1-regulated gene promoters displayed increased levels of divergent transcripts upon Rap1 depletion (Fig 1E). In Sth1-depleted cells, we also observed a notable increase in expression of divergent RNAs in the nascent RNA-seq and, to a lesser extent, in the mRNA-seq (Figs 1E and 2C). The relative changes in divergent RNA transcription after Sth1 and Rap1 co-depletion were comparable to Rap1 depletion, suggesting that at Rap1-regulated genes, in Rap1-depleted cells, RSC does not further promote or repress divergent transcription. As expected, there was little to no effect on divergent transcription of mock-treated samples compared with wild-type (WT) (Fig S1E). These data suggest that RSC depletion

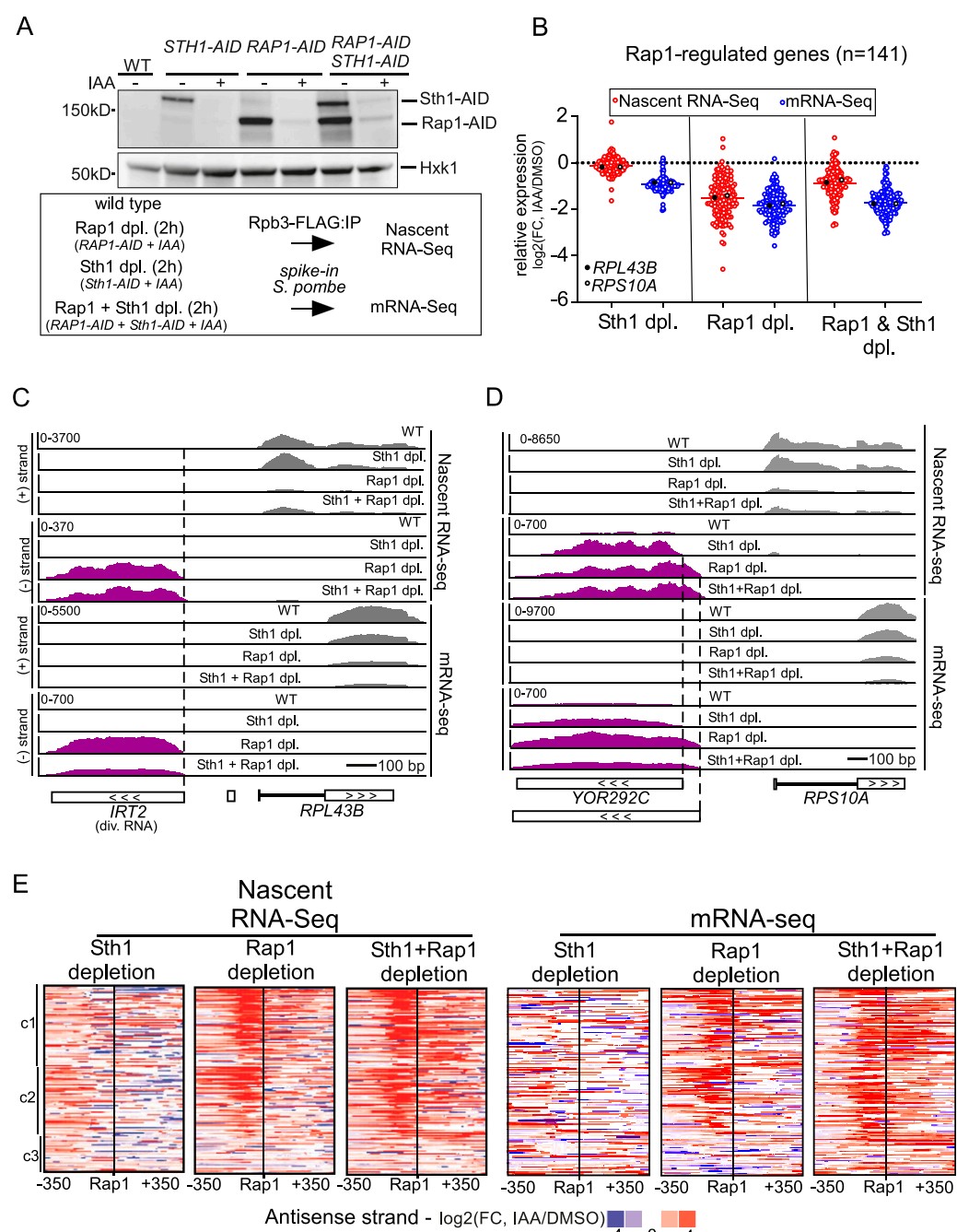

**Figure 1. Depletion of RSC increases divergent transcript accumulation at Rap1-regulated genes.**
**(A)** Top: Rap1 and Sth1 depletions using AID-tagged strains (*RAP1-AID*, *STH1-AID*, *RAP1-AID/STH1-AID*) (FW7238, FW7220, and FW7232) together with WT control (FW7228). Cells were grown until the exponential growth phase and treated with IAA (0.5 mM) for 2 h. AID-tagged proteins were detected by Western blot using anti-V5 antibodies. Hxk1 was used as a loading control, detected using anti-Hxk1 antibodies. One representative biological replicate is shown. Bottom: scheme for nascent RNA-seq and mRNA-seq experiment of Rap1 and Sth1 depletion (dpl). To control for global changes in RNA expression, we added *Schizosaccharomyces pombe* cells as *spike-in* controls (see the Materials and Methods section) and performed mRNA-seq analysis using *S. pombe* (sp) normalisation. **(B)** Differential expression (log$_2$ fold change, FC) of 141 Rap1-regulated genes according to nascent RNA-seq and mRNA-seq, in IAA-treated cells compared with mock-treated cells (DMSO). Each dot represents one coding gene transcript, average values of biological repeats (n = 3). **(C)** Nascent RNA-seq and mRNA-seq signals at the *RPL43B* locus (coding sense direction, grey) and *IRT2* (divergent coding direction, purple). The data of a representative replicate are shown. **(D)** Similar as C, except that, the *RPS10A* locus and the corresponding divergent transcripts are shown. **(E)** Heatmap representing changes in expression levels in the divergent noncoding direction (antisense strand). Promoters were clustered based on antisense strand signal using k-means clustering (k = 3, [c1, c2, and c3]) based on previous analysis for Rap1-regulated gene promoters (Wu et al, 2018). Differences between IAA and DMSO treatment (log$_2$FC) are displayed for each depletion strain, in one representative replicate.
Source data are available for this figure.

causes a relatively greater increase in divergent transcription compared with protein-coding transcription at Rap1-regulated gene promoters, which we further explore in Figs 2–5.

## RSC controls promoter directionality at Rap1-regulated promoters

The promoter directionality score is defined as the ratio between transcription in the sense direction over the divergent direction (Jin et al, 2017) (Fig 2A). We used the directionality score to determine relative changes in the nascent RNA-seq. In short, we computed this metric for all tandem, non-overlapping, and non-divergent gene promoters (n = 2,609) and compared them to Rap1-regulated gene promoters (n = 141) and all genes (n = 6,646) (Figs 2B and S2A). As expected, Rap1-regulated genes are highly directional when compared with all tandem genes (Fig 2B). Upon Rap1 depletion, directionality (as determined by nascent RNA-seq) was strongly reduced because of increased divergent transcription and decreased sense transcription (Fig 2C and D). Also, in Sth1-depleted

cells, the promoter directionality scores were reduced, and divergent transcription increased (Fig 2C and E). By comparing these data with corresponding mRNA-seq data, we identified that Sth1 depletion similarly led to a relative increase in divergent noncoding RNA expression (Fig S2B). Thus, at Rap1-regulated gene promoters, RSC controls promoter directionality by repressing divergent noncoding transcription relative to protein-coding transcription.

## RSC controls promoter directionality genome-wide

Divergent noncoding transcription is an inherent feature of active promoters across species (Jacquier, 2009; Seila et al, 2009; Pelechano & Steinmetz, 2013). Accordingly, many classes of non-coding transcripts have been identified. These include cryptic unstable transcripts (CUTs), which are processed by the nuclear exosome complex, Xrn1-sensitive antisense noncoding RNA (XUTs), Nrd1-unterminated transcripts (NUTs), and stable unannotated transcripts (SUTs) (Neil et al, 2009; Xu et al, 2009; van Dijk et al, 2011; Schulz et al, 2013). These noncoding transcripts (CUTs/SUTs/XUTs/

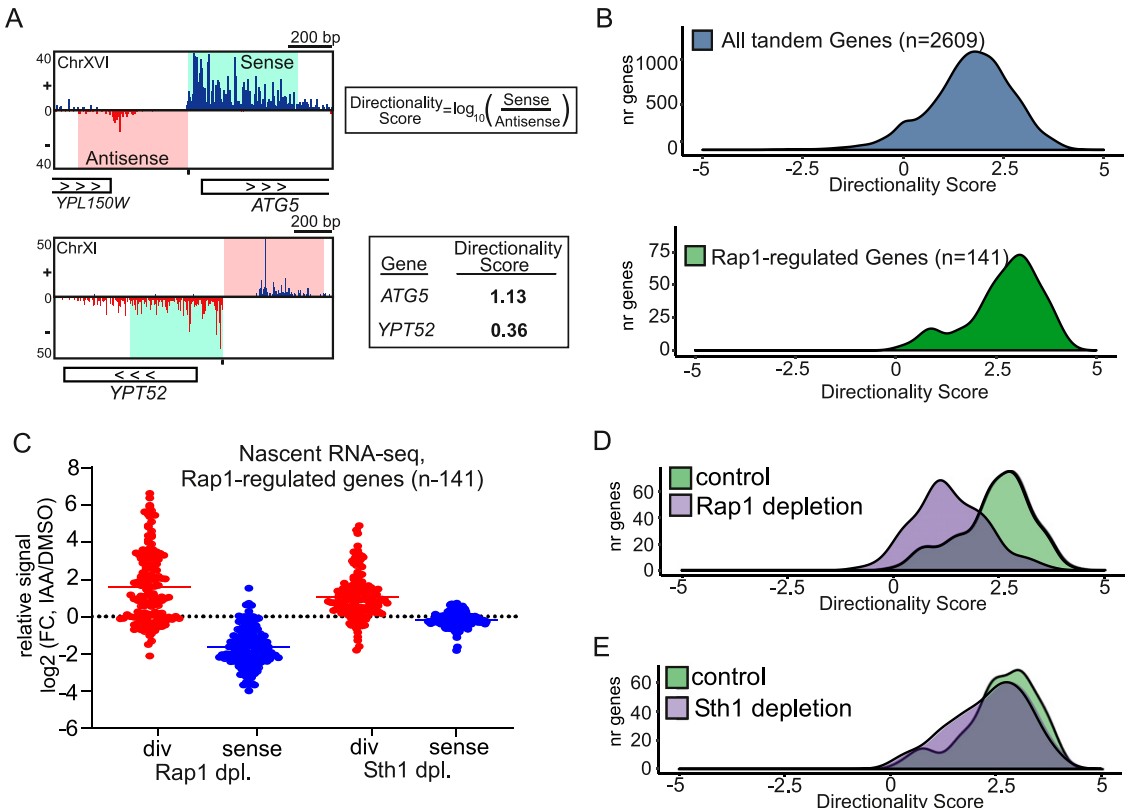

**Figure 2. Depletion of RSC increases divergent transcription and alters promoter directionality.**
**(A)** Approach for calculating directionality score for each promoter. In short, the nascent RNA-seq signals 500 bp upstream (antisense strand) and 500 bp downstream (sense strand) of the coding gene transcription start site were taken. Subsequently, the ratio of sense over antisense signals ($\log_{10}$) was computed, which was defined as the directionality score. Another approach was also described in Jin et al (2017). **(B)** Density plots of directionality scores for tandem non-overlapping genes (n = 2,609) and Rap1-regulated genes (n = 141). **(C)** Relative changes in divergent (red) or sense (blue) transcription levels in cells depleted (dpl) for Rap1 or Sth1 (*RAP1-AID* or STH1-*AID*) in comparison to the corresponding mock-treated cells (IAA/DMSO) for Rap1-regulated gene promoters (n = 141). The nascent RNA-seq signals in the regions upstream (500 bp on the antisense strand, red) and 500 bp downstream (sense strand, blue) of the transcription start site were quantified and compared. Each dot represents one gene, for which the average $\log_2$(FC) value between replicates is shown. **(D)** Density plot representing promoter directionality of control (*RAP1-AID* +DMSO) and Rap1-depleted cells (*RAP1-AID* + IAA) for Rap1-regulated genes (n = 141). Average values between biological replicates (n = 3) are shown. **(E)** Similar as (D), except comparing control (*STH1-AID* + DMSO) and Sth1-depleted (*STH1-AID* + IAA) cells.

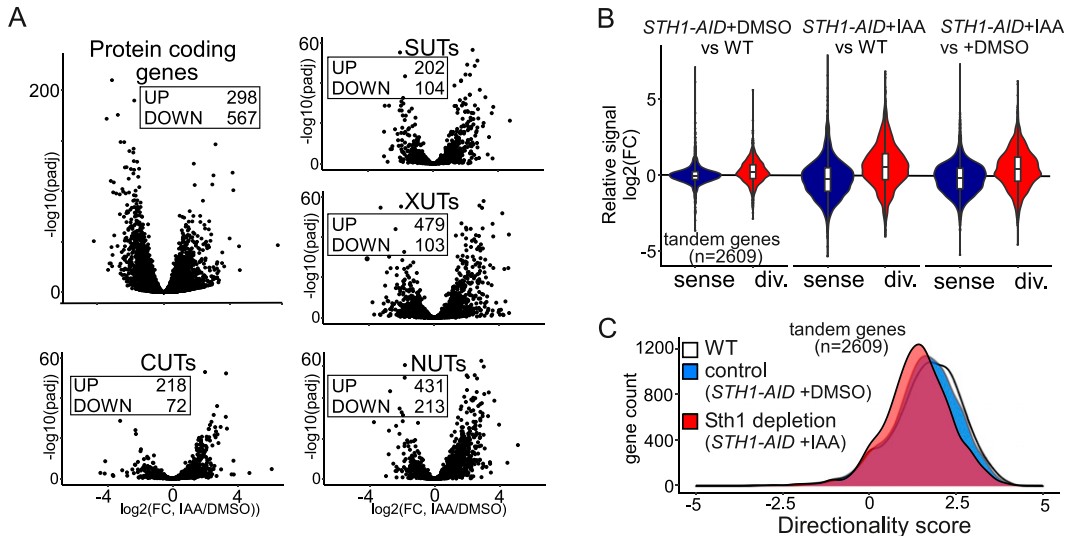

**Figure 3. RSC depletion affects promoter directionality genome-wide.**
**(A)** Volcano plots of nascent RNA-seq data comparing Sth1-depleted to control cells (log₂(FC) IAA/DMSO) for different classes of transcription units. Displayed are plots for protein-coding genes and noncoding transcripts: CUTs, SUTs, XUTs, and NUTs. Numbers of loci with significantly increased (UP) or decreased (DOWN) levels of transcription are displayed (Padj < 0.05 and log₂(FC) > 1). Average values between biological replicates (n = 3) are shown. **(B)** Violin plots displaying the relative changes in divergent (red) or sense (blue) nascent RNA-seq signal of tandem non-overlapping protein-coding genes (n = 2,609). Comparisons between Sth1 depletion (STH1-AID + IAA) and control (STH1-AID + DMSO) or control versus WT are shown. The nascent RNA-seq signals in the regions upstream (500 bp on the antisense strand, red) and 500 bp downstream (sense strand, blue) of the transcription start site were quantified and compared. Average log₂(FC) values between replicates (n = 3) are shown. **(C)** Promoter directionality score of tandem non-overlapping protein-coding genes (n = 2,609) in WT, control (STH1-AID + DMSO), and Sth1-depleted (STH1-AID +IAA) cells. Average values between replicates (n = 3) are represented.

NUTs) are often found at gene promoters and are expressed in the divergent direction, but they are also found at antisense transcription units which overlap protein-coding genes. We examined how expression levels of CUTs/SUTs/XUTs/NUTs were affected upon RSC depletion. Although protein-coding gene expression was relatively decreased (298 genes up-regulated versus 567 genes down-regulated [log₂(FC) > 1, Padj < 0.05]), noncoding transcription of CUTs/SUTs/XUTs/NUTs was substantially up-regulated (1,330 transcripts up-regulated versus 492 transcripts down-regulated [log₂(FC) > 1, Padj < 0.05]) (Fig 3A). Employing the directionality analysis described in Fig 2A, we found that more than 900 promoters (of 2,609 tandem promoters) displayed a relative increase (log₂(FC) > 1) in divergent transcription (Fig 3B and C). Consequently, more than 1,000 gene promoters showed a decrease (log₂(FC) > 1) in directionality score upon Sth1 depletion (STH1-AID + IAA) compared with WT or control cells (STH1-AID + DMSO) (Fig 3C). Notably, control cells showed a decreased directionality score compared with the WT, indicating that the AID tag on Sth1 had itself a small negative impact on RSC function, possibly causing an underestimation of the overall effect of RSC (Figs 3C and S3). We conclude that RSC is important for limiting divergent noncoding transcription relative to protein-coding transcription at least one third of tandem gene promoters.

## RSC limits divergent transcription at promoters with high directionality

To further explore RSC's role in controlling divergent transcription, we examined whether there are features that could explain the effect of RSC on promoter directionality. For example, it is known

that RSC acts on highly transcribed gene promoters and coding sequences (Kubik et al, 2015; Rawal et al, 2018; Biernat et al, 2021). Our data indicate that the ratio of divergent over sense transcription changes upon Sth1 depletion, indicating that RSC's role in promoter directionality is, at least in part, mediated through controlling divergent transcription (Figs 3B and S3). We stratified gene promoters into quintiles (Q1-Q5) based on their directionality score in WT cells, wherein Q5 represents the group of promoters with the highest directionality score and Q1 the lowest. Subsequently, we computed the changes in divergent and sense transcription, as well as changes in directionality score for each quintile upon Sth1 depletion (Fig 4A). Promoters with the highest directionality score in WT cells (Q5) showed the largest relative increase in divergent transcription after Sth1 depletion (Figs 4A and S4A, left panel), suggesting that RSC-mediated repression of divergent transcription is more prominent at highly directional promoters (Figs 4A and S4A, right panel). RSC depletion also affected directionality of gene promoters with the highest sense transcription levels (Q5, sense transcription in WT) (Fig S4B), albeit less compared with the promoters with highest directionality (Q5, directionality score in WT) (Figs 4A and S4C). Control cells (STH1-AID +DMSO) displayed a marginal increase in divergent transcription in the most highly expressed and most directional promoters (Q5 sense transcription and Q5 directionality in WT, respectively) compared with WT cells, which was consistent with the observation that the AID tag partially affected RSC activity and had a small negative impact on promoter directionality (Fig S4D). A similar trend between promoter directionality and RSC-repressed divergent transcription was also observed when we plotted the data using scatter plots

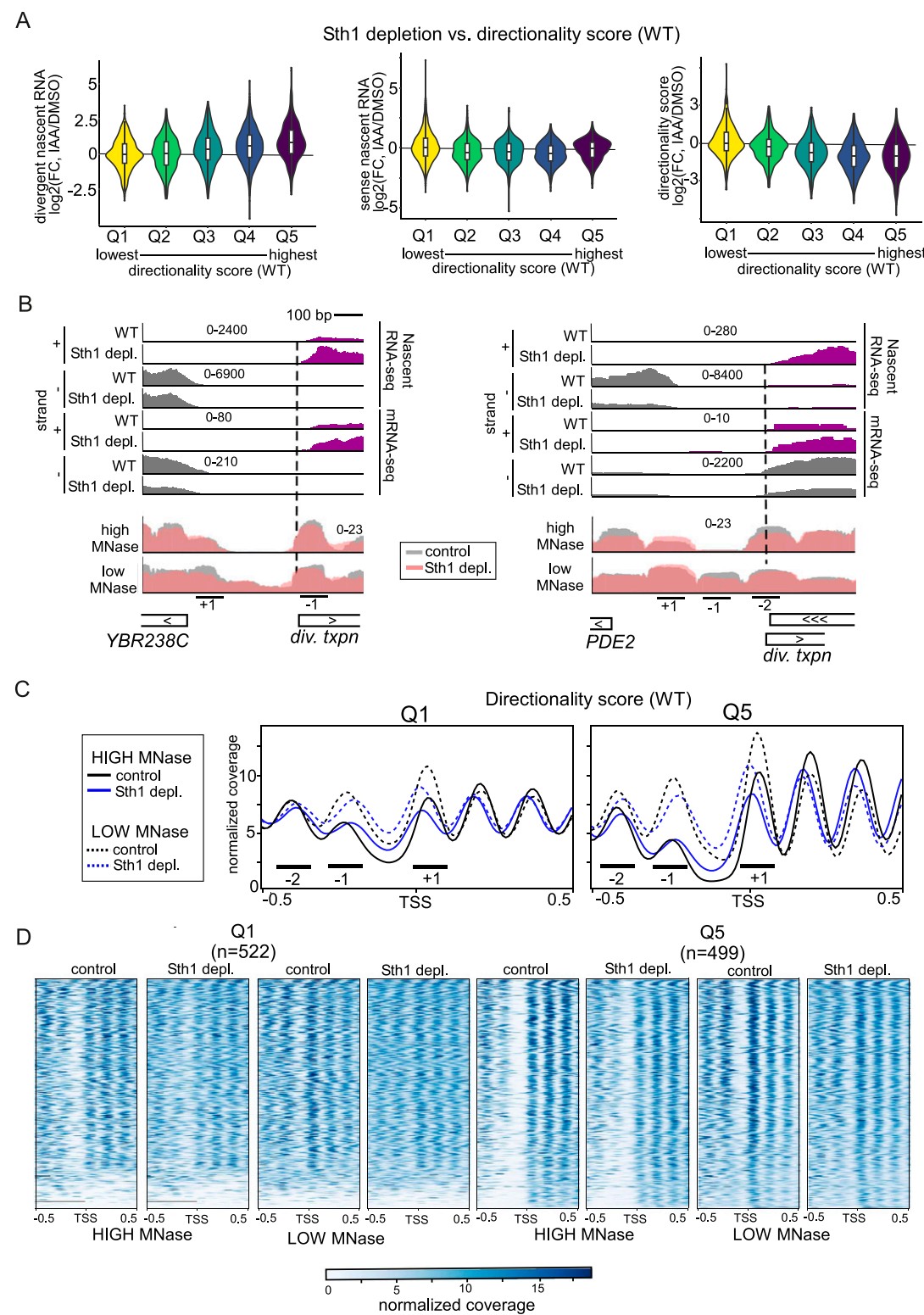

**Figure 4. RSC acts on highly directional promoters.**
**(A)** Violin plots representing the distribution of changes in divergent transcription (left), sense transcription (centre), and directionality score (right) in Sth1-depleted versus control cells (log$_2$(FC) IAA/DMSO), in sets of tandem genes stratified according to their directionality in WT cells (x-axis, Q1 to Q5). The range for directionality scores for each group are the following: Q1(−3.11−0.85), Q2(0.86−1.49), Q3(1.50−1.97), Q4(1.98−2.48), Q5(2.49−4.20). Average values between replicates (n = 3) are represented. **(B)** Examples of loci that display increased divergent transcription and changes in chromatin structure upon Sth1 depletion. The data of a representative replicate of the nascent RNA-seq, mRNA-seq, and MNase-seq are shown. **(A, C)** Metagene analysis of MNase-seq data for Q1 and Q5 (directionality in WT cells) obtained as described in

(Fig S4E and F). Thus, RSC represses divergent over coding gene transcription at promoters that are highly directional.

### RSC mediates nucleosome positioning at highly directional promoters

RSC plays a prominent role in positioning nucleosomes in promoters, which in turn affects PIC recruitment and TSS scanning (Hartley & Madhani, 2009; Klein-Brill et al, 2019). To investigate the mechanism underlying the effects of RSC on promoter directionality, we assessed the chromatin structure in Sth1-depleted cells (Figs 4B–D). Specifically, we compared profiles of MNase sensitivity in cells treated with high and low amounts of MNase using a published dataset (Kubik et al, 2015). When extracts are treated with low concentrations of MNase, partially unwrapped nucleosomes and chromatin-bound non-histone proteins can be detected (Kubik et al, 2015, 2018; Chereji et al, 2017; Yan et al, 2018; Brahma & Henikoff, 2019).

Firstly, we observed the RSC-dependent differences in chromatin structure at individual gene promoters. For example, the *PDE2* and *YBR238C* promoters display increased divergent transcription and a small but notable decrease in MNase-seq signal for the −1 nucleosomes in Sth1-depleted cells (Fig 4B). Notably, the *YBR238C* promoter also harbours a −1 nucleosome (Fig 4B). The broad effect of Sth1 depletion on nucleosome positioning (−2, −1, and +1) in directional promoters may explain why both divergent and sense transcription are often affected upon RSC depletion.

To determine whether the effect at the *PDE2* and *YBR238C* gene promoters reflected genome-wide changes, we analysed nucleosome positioning in the promoter classes sorted by directionality score. Firstly, we found that the group of gene promoters with the highest directionality score (Q5, WT) displayed more defined nucleosome peaks, that were narrower in width at the +1, −1, and −2 positions compared with gene promoters with the lowest directionality score (Q1) (Fig 4C and D, left panel). The −1 and −2 nucleosomes are present in the regions where divergent transcription initiates, whereas the +1 nucleosome is present in the region where transcription initiation from the coding TSS occurs (Fig 4B and C). Secondly, upon depletion of Sth1, nucleosome positioning is broader and less defined for the −1 and −2 positions, whereas the +1 nucleosome shifts more upstream. Thirdly, the +1, −1, and −2 nucleosomes are more sensitive to MNase concentration in Q5 promoters than Q1 promoters (Fig 4C and D, right panel), suggesting a prevalence for partially unwrapped nucleosomes or non-histone proteins in this group. In addition, we inspected another dataset that measured the effect of Sth1 depletion on chromatin using MNase-seq and found that nucleosome positioning was affected in a similar way (compare Fig 4C and D, High MNase, with Fig S4G) (Klein-Brill et al, 2019).

### RSC and GRFs are enriched at highly directional promoters

This work suggests that RSC contributes to limiting relative divergent transcription levels at promoters with high directionality.

One possibility is that RSC is more present at highly directional promoters compared with other promoters. To test this, we analysed RSC binding profiles from a published CUT&RUN dataset (Brahma & Henikoff, 2019). We found that RSC was moderately enriched in highly directional gene promoters (compare Q5 to Q1, Figs 5A and S5A, left panels). We also examined whether DNA sequence motifs associated with RSC binding were more enriched at directional promoters. Both poly-A stretches (A-tracks) and the CG(C/G)G motif are linked with RSC recruitment to promoters (Badis et al, 2008; Lorch et al, 2014; Krietenstein et al, 2016; Kubik et al, 2018). We examined the distribution of these sequences for both the sense and antisense strands. We found that A-tracks and the CG(C/G)G motif were enriched in the group of promoters with the highest directionality (Q5) compared with the group of promoters with lowest directionality (Q1) (Table S1). The distribution of the A-stretches, but not the CG(C/G)G motif, overlapped well with the most enriched regions of RSC binding, suggesting that polyA stretches could be a contributing factor for regulating promoter directionality (Figs 5B and S5B). However, a more detailed analysis is needed to determine the role of RSC binding sites in facilitating RSC recruitment to mediate nucleosome positioning and repression of divergent over coding transcription.

Our data on RSC and Rap1 suggest that both players function in repressing divergent transcription via distinct mechanisms (Wu et al, 2018). Apart from Rap1, other GRFs, such Abf1 and Reb1, are key for controlling promoter architecture and regulating gene transcription and thus possibly play a role in repressing in divergent transcription like Rap1 (Kubik et al, 2017, 2018; Brahma & Henikoff, 2019). We determined whether Abf1 and Reb1 are enriched at directional promoters by analysing a published dataset and comparing groups of gene promoters with the lowest or highest directionality scores (Brahma & Henikoff, 2019). We found that Abf1 and Reb1 were also more enriched for the group of promoters with the highest directionality score (compare Q5 to Q1, Figs 5A and S5A, right panels). This suggests that, apart from Rap1, other GRFs may contribute to repressing divergent transcription, in addition to promoting transcription in the coding direction. To further investigate this hypothesis, we examined a published dataset that measured RNA crosslinked to Pol II (Pol II CRAC) upon Reb1 depletion (Candelli et al, 2018). As expected, we found that depletion of Reb1 increased levels of divergent RNA associated with Pol II, notably at the *BAP2* and *CSG2* promoters (Fig 5C).

### GRFs can repress divergent noncoding transcription when targeted to a divergent promoter

Based on our observations, we predict that Abf1 and Reb1 can repress divergent transcription in promoter regions proximal to their DNA-binding sites (Fig 5A). Previously, we showed that Rap1 represses initiation of divergent transcription near its binding site,

---

(A). Plots are centred on annotated gene transcription start site, and they display MNase-seq signals for chromatin extracts treated with low MNase or high MNase concentrations, obtained from Kubik et al (2015, 2018) (n = 1). Marked are the regions representing the +1, −1 nucleosome, and −2 nucleosome positions. **(C, D)** Heatmap of MNase-seq data represented in (C), centred on the annotated gene transcription start site.

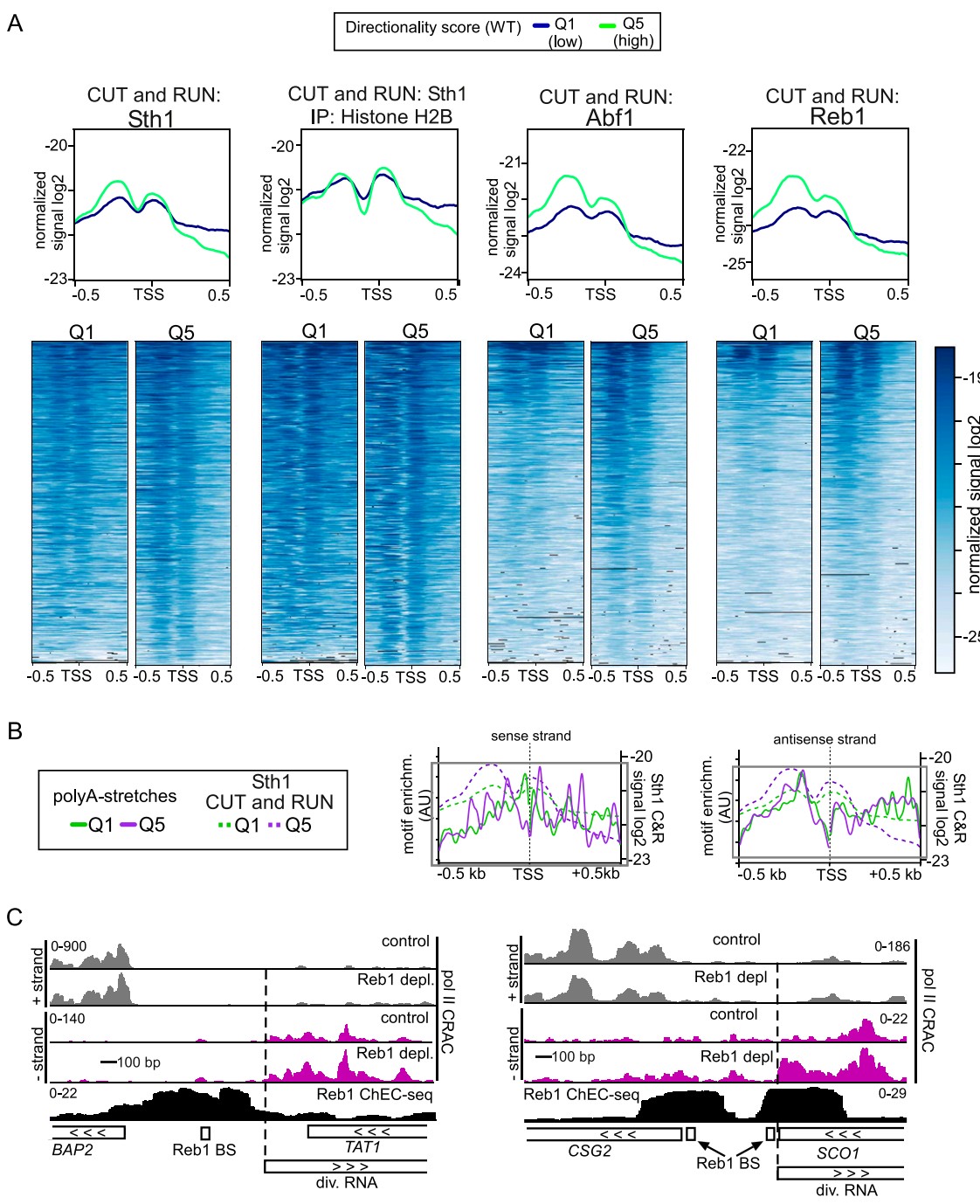

**Figure 5. RSC and general regulatory factors are enriched upstream in promoters with a high directional score.**
**(A)** Metagene analysis and heatmaps of CUT&RUN data of Sth1 (RSC) and the general regulatory factors Abf1 and Reb1. In addition, signals of Sth1 CUT&RUN followed by histone H2B immunoprecipitation are displayed. Data from Brahma and Henikoff (2019). A comparison between genes belonging to directionality Q1 (purple), the group of gene promoters with lowest directionality score, and Q5 (green), the group with the highest directionality score, is shown. The data represent one of n = 2 biological repeats.
**(A, B)** Distribution of the RSC-associated A-track motif for both the sense and antisense strand for Q1 and Q5 overlaid with the RSC CUT&RUN data from panel (A). **(C)** Data from RNA Pol II CRAC and Reb1-ChEC-seq datasets described previously (Zentner et al, 2015; Candelli et al, 2018), representing RNA Pol II transcription in the presence or absence of Reb1 (top) and binding of Reb1 (bottom), respectively. Displayed are the *BAP2* and *SCO1* loci, which show increased divergent transcription upon Reb1 depletion. The positions of the Reb1-binding sites (BS) are indicated.

typically within 50 base pairs (bp) of its binding site (Wu et al, 2018). To functionally test whether indeed Abf1 and Reb1 (and possibly other transcription factors, TFs) repress divergent transcription, we

introduced ectopic transcription factor binding sites adjacent to a divergent core promoter. We used an established fluorescent reporter assay based on the *PPT1* promoter (which normally has a

noncoding transcript *SUT129* in the divergent direction) with both the coding and the divergent noncoding core promoters driving the transcription of fluorescent reporter genes encoding mCherry and YFP, respectively (Marquardt et al, 2014; Wu et al, 2018). We cloned the binding site sequences of several transcription factors 20 bp upstream of the *SUT129*/YFP TSS in the *PPT1* reporter (Fig 6A). We selected GRFs (Cbf1, Abf1, and Reb1) and TFs (Gal4, Gcn4, Cat8, and Gcr1) that have well-defined DNA-binding sequence motifs. To establish if the changes in the reporter signal were dependent on the transcription factor's presence in the cells and not solely because of the alteration in the underlying promoter DNA sequence, we also measured the reporter signal after deleting or depleting the same transcription factors. For example, for the reporter construct with Reb1-binding sites, we measured YFP/mCherry levels in Reb1 control and depleted cells using the auxin-induced degron (*REB1-AID* + IAA), whereas for the reporter construct with Cbf1 binding sites, we measured the YFP/mCherry levels in WT and *cbf1Δ* cells.

Our data indicate that several GRFs were able to repress transcription in the divergent direction and increase promoter directionality (Fig 6B–D). Specifically, *cbf1Δ* cells and depletion of Abf1 and Reb1 (*ABF1-AID* or *REB1-AID* + IAA) decreased the YFP signal (noncoding) compared with control cells and increased promoter directionality (Fig 6B and D). Introduction of binding site motifs for one transcription factor, Gcr1, did not affect noncoding (YFP) direction relative to the WT, but the YFP signal increased in *gcr1Δ* cells, suggesting that Gcr1 also can repress divergent transcription (Fig 6B). The TFs Gal4, Gcn4, and Cat8 had no repressive effect on divergent transcription, perhaps indicating that TFs, unlike GRFs, do not have the ability to repress divergent transcription (Fig 6B). Another possibility is that these three TFs are less active under the growth conditions used in the reporter assay. For example, Cat8 and Gcn4 are most active under non-fermentative growth and amino acid starvation, respectively. For context, Gal4 is bound to promoters under conditions such as growth in rich media (which was tested) (Larschan & Winston, 2001). Interestingly, Gal4 and Gcn4 showed increased signal in the coding direction, suggesting that they are still somewhat active in the conditions tested, but perhaps not sufficiently active for repressing divergent transcription (Fig 6C). We conclude that, as we reported previously for Rap1 (Wu et al, 2018), transcription factors can repress divergent transcription if appropriately positioned adjacent to divergent TSSs.

### Targeting dCas9 to divergent core promoters is sufficient to repress divergent noncoding transcription

Our data indicated that GRFs and RSC repress divergent transcription and thereby promote directionality of promoters (Challal et al, 2018; Wu et al, 2018). One explanation is that these proteins act as barriers for PIC formation and Pol II recruitment and thus prevent transcription initiation. If true, then any protein stably associated with DNA near divergent TSSs should be able to interfere with divergent transcription. To test this, we targeted an exogenous protein to a divergent TSS. We used the catalytically inactivated version of *Streptococcus pyogenes* Cas9 (dCas9), which is a bulky protein (over 150 kD) and has been widely used as tool to modulate

transcription and chromatin (Gilbert et al, 2013). We induced divergent transcription by depleting Rap1 and used gRNAs to target dCas9 to the core promoter of *IRT2*, which is divergently transcribed from *RPL43B* and strongly induced upon Rap1 depletion, Fig 6E–G. As expected, Rap1 depletion induced expression of *IRT2* (Fig 6E and F). Importantly, we did not detect *IRT2* expression when dCas9 was targeted to the *IRT2* TSS (Figs 6H and I and S6). Expression of *IRT2* was still detectable in cells where dCas9 was targeted to the *TEF1* the promoter or upstream in the *MLP1* promoter (*iMLP1*) (Figs 6H and S6). We conclude that steric hindrance by DNA-binding proteins (such as dCas9) is sufficient for repression of divergent transcription.

## Discussion

Transcription from promoters is bidirectional, leading to sense coding and divergent noncoding transcription. However, the directionality (sense over divergent transcription) of promoters greatly varies, indicating that there are mechanisms in place to control directionality of promoters (Jin et al, 2017). Why some promoters are more directional than others is not well understood. Here, we showed that RSC represses divergent noncoding transcription. Our data are consistent with a model where RSC-mediated nucleosome positioning interferes with divergent transcription. Furthermore, we provided evidence that other GRFs (Reb1 and Abf1) and dCas9 are capable of repressing divergent noncoding transcription. We propose that RSC-positioned nucleosomes and other DNA-binding factors form physical barriers in promoters which, in turn, restrict divergent noncoding transcription relative to coding gene transcription.

Because RSC controls global transcription by Pol II, it is important to emphasise the relative nature of the nascent RNA-seq measurement (Parnell et al, 2008). For this reason, we compared the divergent transcription relative to coding gene transcription throughout this manuscript. Noteworthy, the mRNA-seq analysis normalised on external spike-ins showed, albeit less, increased divergent noncoding RNA expression upon RSC depletion. Irrespective of whether the effects of RSC depletion are absolute or relative, evidently divergent noncoding transcription is controlled by RSC in a distinct way compared with coding gene transcription.

### RSC, nucleosome positioning and divergent transcription

The RSC complex, part of the SWI/SNF family of chromatin remodellers, contributes to generating NDRs by "pushing" the nucleosomes positioned at +1 and −1 positions "outwards," thus widening the NDR (Parnell et al, 2008; Hartley & Madhani, 2009; Ganguli et al, 2014). RSC also binds directly to DNA via A-track sequence motifs (Lorch et al, 2014; Krietenstein et al, 2016; Kubik et al, 2018). How does RSC repress divergent transcription? Though the effect was modest, depletion of RSC resulted in less defined consensus nucleosome patterns in the region of promoters where divergent transcription initiates. One interpretation is that the lower nucleosome peaks indicate fewer nucleosomes, which could

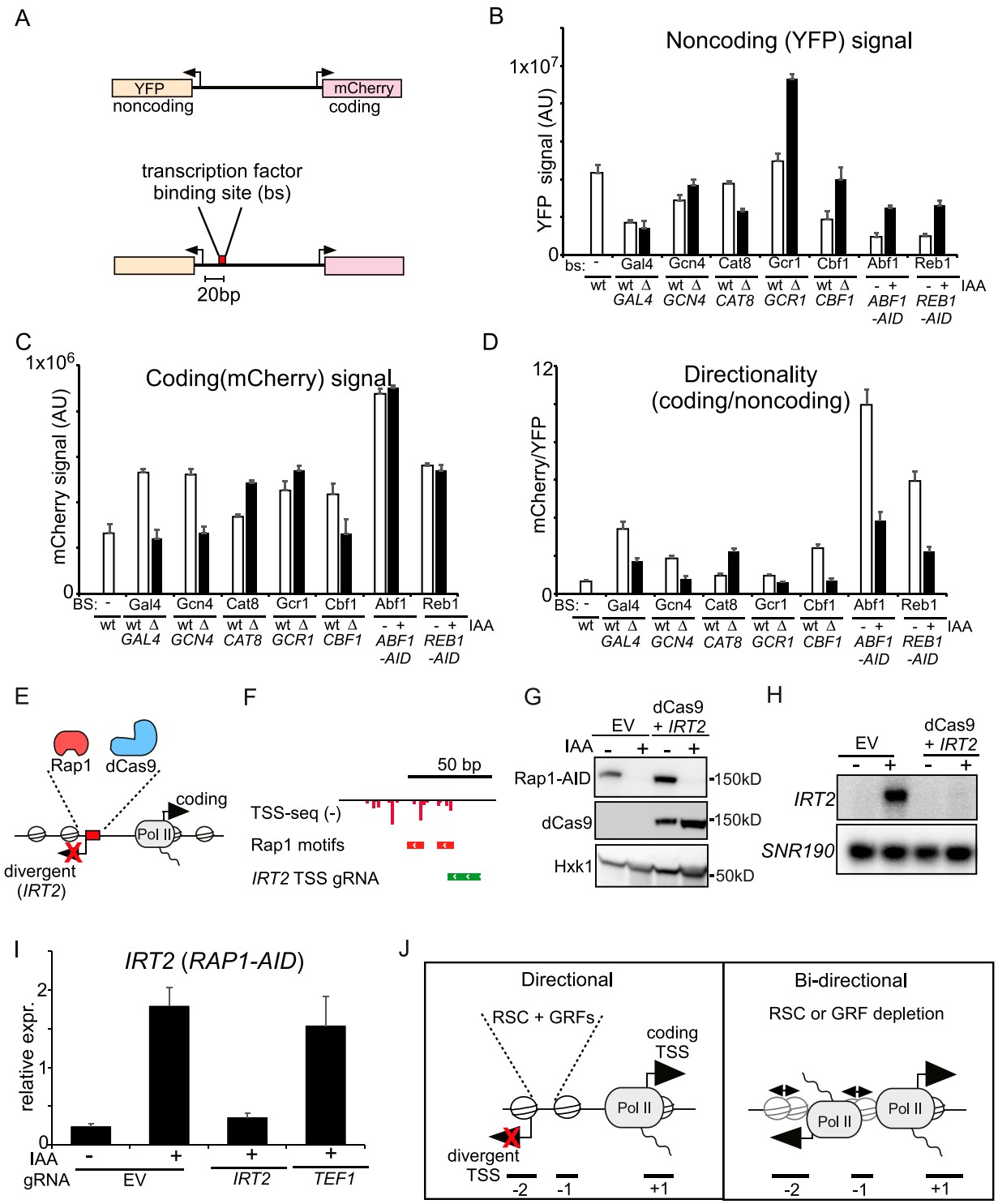

**Figure 6. Targeting general regulatory factors or dCas9 to divergent core promoters is sufficient to repress divergent noncoding transcription.**
**(A)** Schematic overview of the reporter construct. The transcription factor binding sites were cloned 20 nucleotides upstream of the YFP (*SUT129*) transcription start site (TSS). **(B)** YFP, noncoding, signal for constructs harbouring binding sites for Gal4, Gcn4, Cat8, Gcr1, Cbf1, Abf1, and Reb1 cloned into the WT *PPT1* promoter (FW6407). The YFP activity was determined in WT control cells and in gene deletion strains of matching transcription factor binding site reporter constructs (FW6404, FW6306, FW6401, FW6300, FW6402, FW6302, FW6403, FW6424, FW6405, and FW6315). For Abf1 and Reb1 reporter constructs, Abf1 and Reb1 were depleted using the auxin-inducible degron (*ABF1*-AID and *REB1*-AID) (FW6415 and FW6411). Cells were treated for 2 h with IAA or DMSO. Displayed is the mean signal of at least 50 cells. The error bars represent 95% confidence intervals. **(C)** Same analysis as B except that coding direction is shown. **(B, D)** Same analysis as (B) except that directionality was calculated by taking the ratio of coding over noncoding. **(E)** Scheme depicting the targeting of dCas9 to repress the divergent transcript *IRT2*. **(F)** Design of gRNAs targeting the *IRT2* TSS. Displayed are TSS-seq data from after Rap1 depletion (Wu et al, 2018). **(G)** Western blot of Rap1 and dCas9 detected with anti-V5 and anti-FLAG antibodies, respectively. Cells harbouring

allow the transcription machinery to access the DNA and thus result in increased divergent transcription. A more likely explanation is that there could be weaker nucleosome phasing in RSC depleted cells, which flattens the nucleosome peaks and allows access of the transcription machinery to core promoters when the nucleosomes do not occlude the divergent promoter. Interestingly, we found that A-tracks were abundant in promoters on both strands indicating that directional positioning of nucleosomes upstream in promoters may contribute to preventing divergent transcription. Alternatively, RSC could indirectly repress divergent transcription by stimulating coding sense transcription. Indeed, RSC also acts in coding regions to promote Pol II elongation (Biernat et al, 2021). We favour a model where nucleosomes positioned by RSC form barriers to limit PIC formation and Pol II recruitment in the divergent direction (Fig 6J). Upon RSC depletion, the positioning of the −1 and −2 nucleosomes is less strong and PIC formation can occur.

RSC-mediated nucleosome positioning also functions in suppressing other aberrant transcription events. For example, RSC represses antisense transcription initiating from the 3′ ends of gene bodies, which is spatially distinct from the divergent noncoding transcription events we report here (Alcid & Tsukiyama, 2014; Cucinotta et al, 2021). In addition, RSC-mediated positioning of the +1 nucleosome can affect TSS selection, resulting in increased transcription initiation in the sense direction within promoters (Klein-Brill et al, 2019; Kubik et al, 2019; Cucinotta et al, 2021). Thus, RSC has a widespread function in promoting coding gene transcription and limiting aberrant noncoding transcription in the divergent, antisense, and sense directions.

The opening of chromatin controlled by RSC facilitates the recruitment of sequence-specific TFs to promoters (Floer et al, 2010). Therefore it is possible that GRFs or other non-histone proteins mediate the repression of divergent transcription via RSC, as previously proposed (Chereji et al, 2017; Brahma & Henikoff, 2019). Our analysis identified examples of divergent transcription events in which the same promoter was repressed by Rap1 and RSC but using distinct divergent TSSs. This suggests that RSC can repress divergent transcription independently of GRFs (Figs 1D and 6J). How other GRFs act with RSC in controlling divergent transcription remains to be dissected further. Based on our analysis, it appears more likely that RSC and GRFs act together to repress divergent transcription and promote transcription in the protein-coding direction.

### Model for promoter directionality

GRFs and positioned nucleosomes constitute essential components for promoter organisation. We propose that these proteins physically interfere with divergent transcription when bound to DNA (Fig 6J). Indeed, deposition of nucleosomes by chromatin assembly factor I (CAF-I) plays a widespread role in repressing divergent transcription (Marquardt et al, 2014). Here, we showed that RSC is important to position nucleosomes and to ensure that cryptic divergent promoters are "protected" and transcriptionally repressed. Conversely, opening of chromatin because of increased histone lysine 56 acetylation (H3K56ac) leads to more divergent transcription in yeast, whereas deacetylation of histone H3 by Hda1 deacetylase complex (Hda1C) facilitates repression of divergent transcription (Marquardt et al, 2014; Gowthaman et al, 2021 Preprint). The GRF Rap1 can repress divergent transcription at gene promoters by occupying cryptic divergent promoters (Wu et al, 2018). We showed that Abf1 and Reb1 have potentially the same ability. In addition, dCas9 has the same capacity to repress aberrant noncoding transcription when targeted to divergent core promoter. Also, sequence-specific DNA-binding factors, such as Tbf1 and Mcm1, have the ability to insulate two independently regulated divergent gene promoter pairs from each other and thus effectively repress divergent transcription (Yan et al, 2015).

In mammalian cells, the multifunctional transcription factor CTCF directly represses initiation of divergent noncoding transcription at hundreds of promoters, indicating that the mechanism of DNA-binding factors repressing divergent transcription is likely conserved and widespread (Luan et al, 2021 Preprint). The RSC related mammalian esBAF complex has comparable functions in repressing noncoding transcription to some extent, suggesting that the role of RSC in controlling divergent transcription and promoter directionality is also likely conserved (Mohrmann & Verrijzer, 2005; Hainer et al, 2015). Like RSC, esBAF positions the nucleosomes flanking the NDR. Depletion of esBAF leads to less stable nucleosome positioning, and consequently to widespread increased noncoding transcription, of which some are divergent noncoding transcription events.

Further work will be required to dissect the interplay between GRFs and chromatin states in controlling aberrant noncoding transcription and promoter directionality. These regulatory complexes are present and conserved across eukaryotes. Thus, our study may provide important insights on how promoter directionality is controlled in multicellular organisms.

## Materials and Methods

### Strains, plasmids and growth conditions

Strains isogenic to the *S. cerevisiae* BY4741 strain background (derived from S288C) were used for this investigation. A one-step

*RAP1*-AID were treated with IAA to deplete Rap1 and induce *IRT2* expression in either cells with empty vector (EV) or cells harbouring a dCas9 construct and a construct expressing the gRNA targeted to the Rap1-binding site (FW8477, FW8531). Hxk1 was used as a loading control, detected using anti-Hxk1 antibodies. **(E, H)** *IRT2* expression detected by Northern blot for strains and treatments described in (E). As a loading control, the membrane was re-probed for *SNR190*. **(D, I)** *IRT2* expression as detected in by qRT-PCR of as described in (D). A control gRNA was included in the analysis targeted to the TSS of *TEF1* (FW8527). Displayed are the mean signals of n = 3 biological repeats and SEM. **(J)** Model for promoter directionality.
Source data are available for this figure.

tagging procedure was used to generate endogenous carboxy (C)-terminally tagged auxin-inducible degron (AID) alleles of Rap1 and Sth1 (Nishimura et al, 2009). The *RAP1-AID* strain was previously described and contains three copies of the V5 epitope and the IAA7 degron (Wu et al, 2018). For tagging *RPB3* with the FLAG epitope, we used a one-step tagging procedure using a plasmid (3xFLAG-pTEF-NATMX-tADH1, gift from Jesper Svejstrup). The single-copy integration plasmids for expression of *Oryza sativa TIR1* (*osTIR1*) ubiquitin E3 ligase were described previously (Wu et al, 2018).

For auxin-inducible degron (*AID*) depletion experiments, induction of AID-tagged protein depletion was performed with 3-indole-acetic acid (IAA, 0.5 mM) (Sigma-Aldrich) during the exponential growth phase (approximately $OD_{600}$ 0.8).

The *pPPT1-pSUT129* (*pPS*, mCherry-YFP) fluorescent reporter plasmid was described previously (gift from Sebastian Marquardt, University of Copenhagen) (Marquardt et al, 2014). The target motifs for each transcription factor (obtained from the YeTFaSCo database [de Boer & Hughes, 2012]) were introduced by blunt-end cloning into unique restriction site (SspI) proximal to the *SUT129* TSS and verified by Sanger sequencing. After linearisation by digestion with EcoRI, the reporter construct was transformed and integrated to replace the endogenous *PPT1-SUT129* locus.

For CRISPR interference (CRISPRi) experiments, yeast expression constructs for nuclease-inactivated Cas9 (D10A/H840A mutations) from *S. pyogenes*–dCas9 (#46920; Addgene) (Gilbert et al, 2013; Qi et al, 2013) were sub-cloned into single-copy integration plasmids by restriction cloning. The 3xFLAG epitope was introduced by Gibson-style cloning (NEBuilder HiFi, NEB) at the C-terminus of the construct, in-frame with the ORF. dCas9-3xFLAG expression construct on single-copy integration plasmids was transformed into yeast after linearisation with PmeI. Single-guide RNA (sgRNA) plasmids (gift from Elçin Ünal, UC Berkeley) (Chen et al, 2017) were generated by site-directed mutagenesis (Q5 Site-Directed Mutagenesis Kit, NEB), using specific primers containing the sgRNA 20-mer target sequence (split in half across the two primers). sgRNA target sequence selection was guided by availability of "NGG" protospacer adjacent motif (PAM) sites and transcription start-site sequencing (TSS-seq) data (Wu et al, 2018). sgRNA expression plasmids were integrated at the *SNR52* locus after linearisation by XbaI digestion. Strains, plasmids, and oligonucleotide sequences are listed in Tables S2–S4, respectively.

### RNA extraction

Yeast cells were harvested from cultures for RNA extraction by centrifugation and then washed once with sterile water before snap-freezing in liquid nitrogen. RNA was extracted from frozen yeast cell pellets using acid phenol:chloroform:isoamyl alcohol (125:24:1, Ambion) and Tris–EDTA-SDS (TES) buffer (0.01 M Tris–HCl, pH 7.5, 0.01 M EDTA, 0.5% wt/vol SDS), by rapid agitation (1,400 rpm,

65°C for 45 min). After centrifugation, the aqueous phase was collected, and RNA was precipitated at −20°C overnight in ethanol with 0.3 M sodium acetate. After centrifugation and washing with 80% (vol/vol) ethanol solution, dried RNA pellets were resuspended in DEPC-treated sterile water and subsequently stored at −80°C.

### qRT-PCR

Reverse transcription and quantitative PCR for *IRT2* performed as described previously (Moretto et al, 2018; Tam & van, Werven, 2020). In short, total RNA was purified using the NucleoSpin RNA kit (Macherey-Nagel) according to the manufacturer's instructions. For reverse transcription, the ProtoScript II First Strand cDNA Synthesis Kit (New England BioLabs) was used and 500 ng of total RNA was provided as template in each reaction. qRT-PCR reactions were prepared using EXPRESS SYBR GreenER SuperMix (Thermo Fisher Scientific), and *IRT2* levels were quantified on the Applied Biosystems 7500 Fast Real-Time PCR System (Thermo Fisher Scientific). The result of n = 3 biological replicates are shown. Signals were normalised over *ACT1*. Primer sequences are listed in Table S4.

### Western blot

Western blots were performed as described previously (Chia et al, 2017; Chiu et al, 2018). Protein extracts were prepared from whole cells after fixation with trichloroacetic acid (TCA). Samples were pelleted by centrifugation and incubated with 5% wt/vol TCA solution at 4°C for at least 10 min. Samples were washed with acetone, pelleted, and dried. Samples were then resuspended in protein breakage buffer (50 mM Tris [pH 7.5], 1 mM EDTA, 2.75 mM DTT) and subjected to disruption using a Mini Beadbeater (Biospec) and 0.5-mm glass beads. Protein extract samples were mixed with SDS–PAGE sample buffer (187.5 mM Tris [pH 6.8], 6.0% vol/vol β-mercaptoethanol, 30% vol/vol glycerol, 9.0% vol/vol SDS, 0.05% wt/vol Bromophenol blue) in a 2:1 ratio by volume, and protein samples were denatured at 95°C for 5 min.

SDS–PAGE (polyacrylamide gel electrophoresis) was performed using 4–20% gradient gels (Bio-Rad TGX), and samples were then transferred onto PVDF membranes by electrophoresis (wet transfer in cold transfer buffer: 3.35% wt/vol Tris, 14.9% wt/vol glycine, 20% vol/vol methanol). Membranes were incubated in blocking buffer (1% wt/vol BSA, 1% wt/vol nonfat powdered milk in phosphate-buffered saline with 0.01% vol/vol Tween-20 [PBST] buffer) before primary antibodies were added to blocking buffer for overnight incubation at 4°C. For probing with secondary antibodies, membranes were washed in PBST and anti-mouse or anti-rabbit IgG HRP-linked antibodies were used for incubation in blocking buffer (1 h, room temperature). Signals corresponding to protein levels were detected using Amersham ECL Prime detection reagent and an Amersham Imager 600 instrument (GE Healthcare).

## Antibodies

**The following antibodies were used for Western blotting.**

| Antibody | Dilution | Source | Reference |
|---|---|---|---|
| Anti-hexokinase rabbit IgG | 1:8,000 | US Biological | H2035 |
| Anti-FLAG mouse monoclonal IgG$_1$, clone M2 | 1:2,000 | Sigma-Aldrich (Merck) | F3165 |
| Amersham ECL anti-mouse IgG, HRP-linked whole antibody (from sheep) | 1:10,000 | GE Life Sciences | NA931V5 |
| Amersham ECL anti-rabbit IgG, HRP-linked whole antibody (from donkey) | 1:10,000 | GE Life Sciences | NA934V |

### Northern blot

Northern blots were performed as described previously (Chia et al, 2017; Wu et al, 2018). Briefly, RNA samples (10 μg per lane) were incubated in sample denaturation buffer (1 M deionised glyoxal, 50% vol/vol DMSO, 10 mM sodium phosphate [NaP$_i$] buffer [pH 6.8]) at 70°C for 10 min, loading buffer (10% vol/vol glycerol, 2 mM NaP$_i$ buffer, 0.4% wt/vol bromophenol blue) was added, and RNA samples were subjected to electrophoresis (2 h at 80 V) on an agarose gel (1.1% vol/vol agarose, 0.01 M NaP$_i$ buffer). Capillary transfer was used to transfer total RNA onto positively charged nylon membranes (GE Amersham Hybond N+). Bands corresponding to mature rRNA were visualised by staining with methylene blue solution (0.02% wt/vol methylene blue, 0.3 M sodium acetate).

The nylon membranes were incubated for at least 3 h at 42°C in hybridisation buffer (1% wt/vol SDS, 40% vol/vol deionised form-amide, 25% wt/vol dextran sulphate, 58 g/l NaCl, 200 mg/l sonicated salmon sperm DNA [Agilent], 2 g/l BSA, 2 g/l polyvinyl-pyrolidone, 2 g/l Ficoll 400, 1.7 g/l pyrophosphate, 50 mM Tris, pH 7.5) or ULTRAhyb Ultrasensitive Hybridization Buffer (Thermo Fisher Scientific) to minimise non-specific probe hybridisation. Probes were synthesised using a Prime-it II Random Primer Labelling Kit (Agilent), 25 ng of target-specific DNA template, and radioactively labelled with dATP (α-32P) (Perkin-Elmer or Hartmann Analytic). The oligonucleotides used to amplify target-specific DNA templates for *IRT2* and *SNR190* Northern blot probes by PCR can be found in Table S4.

Blots were hybridised overnight with radioactively labelled probes at 42°C and then washed at 65°C for 30 min each with the following buffers: 2× saline-sodium citrate (SSC) buffer, 2× SSC with 1% wt/vol SDS, 1× SSC with 1% SDS, and 0.5× SSC with 1% SDS. For image acquisition, membranes were exposed to storage phosphor screens before scanning on the Typhoon FLA 9500 system (GE Healthcare Life Sciences). To strip membranes before re-probing for different transcripts, membranes were washed with stripping buffer (1 mM Tris, 0.1 mM EDTA, 0.5% wt/vol SDS) at 85°C until negligible residual signal remained on the blots.

### Fluorescence microscopy

Yeast cells were grown in YPD media (small batch cultures) to the exponential growth phase and fixed with formaldehyde (3.7% wt/vol), incubating at room temperature for 15 min. Fixed cells were washed with phosphate-sorbitol buffer (0.1 M KP$_i$ [pH 7], 0.05 M MgCl$_2$, 1.2 M sorbitol) and resuspended in the same buffer before imaging. Images were acquired using a Nikon Eclipse Ti-E inverted microscope imaging system (Nikon) equipped with a 100× oil objective (NA 1.4), SOLA SE light engine (Lumencor), ORCA-FLASH 4.0 camera (Hamamatsu), and NIS-Elements AR software (Nikon). 500-ms exposure time was specified and GFP, and mCherry filters were used to detect YFP and mCherry signals, respectively.

To quantify whole-cell fluorescence signals, measurements were performed using ImageJ (version 1.52i, NIH) (Schneider et al, 2012) for the YFP and mCherry channels. Regions of interest were manually drawn around the border of each cell. Mean signal represents the mean intensity in each channel per cell multiplied by the cell area, and the signal for YFP and mCherry was corrected for cell-free background fluorescence in a similar manner. WT cells without integrated fluorescent reporter plasmids were also measured to determine auto-fluorescence signal. 50 cells were quantified for each sample.

### RNA-seq with *S. pombe* spike-ins (mRNA-seq)

Total RNA was extracted from *S. cerevisiae* pellets spiked-in with *S. pombe* cells in a 10:1 ratio of *S. cerevisiae*:*S. pombe* as described above, treated with rDNase in solution (Machery-Nagel), and purified by spin column (Machery-Nagel) before preparation of sequencing libraries. 500 ng of RNA input material was used for polyadenylated RNA (mRNA) sequencing. Libraries were prepared using the TruSeq Stranded mRNA kit (Illumina) according to the manufacturer's instructions (10 or 13 PCR cycles). The libraries were multiplexed and sequenced on either the HiSeq 2500 or 4000 platform (Illumina) and generated ~45 million 101-bp strand-specific paired-end reads per sample on average.

### Nascent RNA sequencing (nascent RNA-seq)

For nascent RNA sequencing (nascent RNA-seq), RNA fragments associated with RNA polymerase II subunit Rpb3 endogenously tagged with 3xFLAG epitope at the C-terminus were isolated by affinity purification as described previously (Churchman & Weissman, 2011, 2012; Moretto et al, 2021). Small batch cultures

of yeast cells grown in YPD media were collected by centrifugation, the supernatant was aspirated, and cell pellets were immediately snap-frozen by submerging in liquid nitrogen to minimise changes in nascent transcription activity (e.g., in response to cell resuspension in cold lysis buffer with high concentrations of salts and detergents). Frozen cell pellets were dislodged from centrifuge tubes and stored at –80°C. Cells were subjected to cryogenic lysis by freezer mill grinding under liquid nitrogen (SPEX 6875D Freezer/Mill, standard program: 15 cps for six cycles of 2 min grinding and 2 min cooling each). Yeast "grindate" powder was stored at –80°C. 2 g of yeast grindate was resuspended in 10 ml of 1× cold lysis buffer (20 mM Hepes, pH 7.4, 110 mM potassium acetate, 0.5% vol/vol Triton-X-100, 1% vol/vol Tween-20) supplemented with 10 mM $MnCl_2$, 1× Roche cOmplete EDTA-free protease inhibitor, and 50 U/ml SUPERase.In RNase Inhibitor. Chromatin-bound proteins were solubilised by incubation with 1,320 U of Dnase I (RQ1 Rnase-free Dnase I; Promega) on ice for 20 min. Lysates containing solubilised chromatin proteins were clarified by centrifugation at 20,000$g$ for 10 min (at 4°C), and the supernatant was taken as input for immunoprecipitation using 500 $\mu$l of anti-FLAG M2 affinity gel suspension (A2220; Sigma-Aldrich) per sample (2.5 h at 4°C). After immunoprecipitation, the supernatant was removed, and beads were washed four times with 10 ml of cold wash buffer each time (1× lysis buffer with 50 U/ml Superase.In Rnase inhibitor and 1 mM EDTA). After the last wash, agarose beads were transferred to small chromatography spin columns (Pierce Spin Columns; Thermo Fisher Scientific), and competitive elution of protein complexes containing Rpb3-3xFLAG protein from the resin was performed by incubating beads with 300 $\mu$l of elution buffer (1× cold lysis buffer with 2 mg/ml 3xFLAG peptide) for 30 min at 4°C (3xFLAG peptide provided by the Peptide Chemistry Science Technology Platform; the Francis Crick Institute). Elution was performed twice and 600 $\mu$l of eluate was subjected to acid phenol:chloroform RNA extraction and ethanol precipitation. A significant amount of 3xFLAG peptide co-precipitates with the RNA as a contaminant and is later removed by spin column purification.

Purified RNA was fragmented to a mode length of ~200 nucleotides using zinc ion–mediated fragmentation (AM870; Ambion, 70°C for 4 min). Fragmented RNA was purified using miRNeasy spin columns (miRNeasy mini kit; QIAGEN), which retain RNAs ~18 nucleotides or more in length. Purified RNA was quantified by Qubit (Thermo Fisher Scientific), and ~150 ng of RNA was subjected to rRNA depletion using the Ribo-Zero Gold rRNA Removal Kit (Yeast) (MRZY1324; Illumina, now discontinued). Libraries were prepared using the TruSeq Stranded Total RNA kit (Illumina) according to the manufacturer's instructions (14 PCR cycles). The libraries were multiplexed and sequenced on the HiSeq 4000 platform (Illumina) and generated ~28 million 101-bp strand-specific paired-end reads per sample on average.

## RNA-seq data analysis

Adaptor trimming was performed with cutadapt (version 1.9.1) (Martin, 2011) with parameters "–minimum-length = 25 –quality-cutoff = 20 -a AGATCGGAAGAGC -A AGATCGGAAGAGC." BWA (version 0.5.9-r16) (Liao et al, 2014) using default parameters was used to perform the read mapping independently to both the *S. cerevisiae* (assembly R64-1-1, release 90) and *S. pombe* (assembly ASM294v2, release 44) genomes. Genomic alignments were filtered to only include those that were primary, properly paired, uniquely mapped, not soft-clipped, maximum insert size of 2 kb, and fewer than three mismatches using BamTools (version 2.4.0; [Barnett et al, 2011]). Read counts relative to protein-coding genes were obtained using the featureCounts tool from the Subread package (version 1.5.1) (Liao et al, 2014). The parameters used were "-O –minOverlap 1 –nonSplitOnly– primary -s 2 -p -B -P -d 0 -D 1000 -C –donotsort."

Differential expression analysis was performed with the DESeq2 package (version 1.12.3) within the R programming environment (version 3.3.1) (Love et al, 2014). The spiked-in *S. pombe* transcripts were used to assess the size factors, potentially mitigating any impact a global shift in the *S. cerevisiae* read counts would have on DESeq2's usual normalisation procedure. PCA was performed with DESeq2 on 500 genes with the highest variance.

### Nascent RNA-seq data analysis

Adaptor trimming was performed with cutadapt (version 1.9.1) (Martin, 2011) with parameters "–minimum-length = 25 –quality-cutoff = 20 -a AGATCGGAAGAGC -A AGATCGGAAGAGC -u –50 -U –50." The RSEM package (version 1.3.0) (Li & Dewey, 2011) in conjunction with the STAR alignment algorithm (version 2.5.2a) (Dobin et al, 2013) was used for the mapping and subsequent gene-level counting of the sequenced reads with respect to all *S. cerevisiae* genes downloaded from the Ensembl genome browser (assembly R64-1-1, release 90; [Kersey et al, 2016]). The parameters used were "–star-output-genome-bam–forward-prob 0," and all other parameters were kept as default. Differential expression analysis was performed with the DESeq2 package (version 1.12.3) (Love et al, 2014) within the R programming environment (version 3.3.1). An adjusted *P*-value of ≤ 0.01 was used as the significance threshold for the identification of differentially expressed genes.

STAR genomic alignments were filtered to include only unspliced, primary, uniquely mapped, and properly paired alignments with a maximum insert size of 500 bp. The featureCounts tool from the Subread package (version 1.5.1) (Liao et al, 2014) was used to obtain insert fragment counts within defined intervals (e.g., +100 bp) around the 141 Rap1-binding sites (separate intervals for Watson-and-Crick strand alignments, 242 intervals total) using the parameters "-O –minOverlap 1 –nonSplitOnly–primary -s 2 -p -B -P -d 0 -D 600 -C." Strand-specific reads were only counted if they overlapped with the interval on the corresponding strand. DESeq2 was used to perform differential expression analysis for intervals around Rap1-binding sites as described above, and the DESeq2 size factors calculated with respect to the transcriptome or *S. pombe* transcriptome were used for normalisation of the per-sample counts. Genomic annotations of CUTs, SUTs, XUTs, and NUTs (various noncoding RNA species) was obtained from Wery et al (2016) and filtered for transcripts greater than 200 nt in length to match the insert size distribution of RNA-seq libraries. As for mRNA-seq, PCA was performed with DESeq2 on 500 genes with the highest variance.

## MNase-seq and CUT&RUN analysis

Publicly available datasets were obtained from NCBI Gene Expression Omnibus (GEO) MNase-seq (GSE73337, GSE98260) (Kubik et al, 2015; Challal et al, 2018). Adaptors were trimmed from MNase-seq reads using cutadapt as described above. Adaptor-trimmed reads were mapped to the *S. cerevisiae* genome (Ensembl assembly R64-1-1, release 90) (Zerbino et al, 2018) with BWA (version 0.5.9-r16) (Li & Durbin, 2009) using default parameters. Only uniquely mapped and properly paired alignments that had no more than two mismatches in either read of the pair and an insert size 120–200 bp were kept for paired-end Mnase-seq alignments. To generate genome-wide coverage tracks for nucleosome occupancy from MNase-seq data, the DANPOS2 dpos command (version 2.2.2) (Chen et al, 2013) was used with parameters "–span 1 –smooth_width 20 –width 40 –count 1000000."

CUT&RUN data were obtained from GSE116853 (Brahma & Henikoff, 2019). The binding of Sth1/Abf1/Reb1 from the above-mentioned CUT&RUN-ChIP data were plotted for genes belonging to the specified categories with deepTools (version 3.3.0) (Ramirez et al, 2016), using the computeMatrix parameters "reference-point–referencePoint TSS–upstream 500 –downstream 500 –skipZeros." The bedgraph files of CUT&RUN-ChIP data were previously clipped to the size of sacCer3 chromosomes, and values were normalised over the sum of each sample and transformed to log$_2$.

## Promoter directionality score analysis

To calculate directionality scores for coding gene promoters, a curated list of coding gene TSSs was obtained from published TSS sequencing data as described (Park et al, 2014). Any missing TSS coordinates for coding genes were supplemented with the TSS annotation from Ensembl (assembly R64-1-1, release 90) (Zerbino et al, 2018), generating a list of 6,646 *S. cerevisiae* coding gene TSSs. To avoid quantification of divergent direction transcription that constituted coding transcription for an upstream divergent gene, overlapping and divergent gene pairs were removed from the analysis resulting in 2,609 non-overlapping and tandem genes. To simplify the counting analysis, the coverage for each paired-end read from nascent RNA-seq was reduced to the single 3′ terminal nucleotide of the strand-specific read using genomeCoverageBed function within BEDTools (version 2.26.0) (Quinlan & Hall, 2010) with parameters "-ibam stdin -bg -5 -scale %s -strand %s." "Sense" direction windows encompassed nucleotide positions +1 to +500 in the coding direction relative to the TSS and "antisense" direction windows encompassed nucleotide positions –1 to –500 in the divergent direction relative to the TSS. The total number of reads with 3′ end positions falling within the sense and antisense direction windows was quantified for each gene using the computeMatrix tool within deepTools (version 2.5.3) (Ramirez et al, 2016) using parameters "reference-point–referencePoint center–upstream 500 –downstream 500 –binSize 1 –scale 1." A value of one was added to all counts to avoid dividing by zero. For each window, the mean read count was calculated from three biological replicate experiments and used for subsequent calculation and plotting. For divergent and sense signals, the sum of the signal in the 500 nt upstream (antisense) and downstream (sense) windows, respectively, was calculated for every gene, and ratios were computed and compared between samples.

## Stratification of promoters and sequence analysis

Gene promoters were stratified in quintiles (Q1 to Q5) using Matt (version 1.3.0) (Gohr & Irimia, 2019), according to sense transcription levels or directionality score (log$_{10}$) in the WT strain. Analyses were repeated on only promoters with intermediate gene transcription levels (belonging to Q3 according to sense transcription level). Analysis for enrichment and distribution of A-tracks (AAAAAAA) and GC-rich motifs (CG(C/G)G) as defined in Kubik et al (2015) was performed with Matt (version 1.3.0), using the *test_regexp_enrich* and *get_regexp_prof* functions (Gohr & Irimia, 2019).

## Data plotting and visualisation

Bar plots, scatter plots, and volcano plots were generated using GraphPad Prism (version 7 or 8). Screenshots of sequencing data were captured using the Integrative Genomics Viewer (IGV, Broad Institute, version 2.4.15) (Robinson et al, 2011). The RStudio integrated development environment (version 1.0.143) was used within the R statistical computing environment (version 3.4.0) for data analysis and visualisation. Software packages within the tidyverse (version 1.2.1) collection were used for data analysis and plotting. The following functions within ggplot (version 3.0.0) were used for plotting: violin plots, geom_violin with parameters "scale = count"; geom_boxplot with parameters "outlier.colour = NA"; smoothed density plots, geom_density with default parameters; scatter plots, geom_point with default parameters; marginal density histogram plots, ggMarginal with parameters "type = "histogram," bins = 40, size = 8. The Cairo graphics library (version 1.17.2) was used to generate heatmap plots for the RNA-seq data after fold change values were calculated for bins of 5 nt within defined intervals, comparing between two samples.

## Statistical analysis

Information regarding any statistical tests used, number of samples, or number of biological replicate experiments is stated in the corresponding figure legends. For *t* tests, calculated *P*-values less than 0.05 were considered significant. Plotted error bars in individual figures are stated in the figure legend, as either SEM or 95% confidence intervals (CI).

## Publicly available datasets used in this study

MNase-seq data were obtained from GSE73337 (Kubik et al, 2015) and GSE98260 (Kubik et al, 2018), respectively. TSS annotation was described in GSE49026 (Park et al, 2014). Rap1-regulated gene promoters were described in Wu et al (2018). The annotations of CUTs, SUTs, NUTs, and XUTs were described in Neil et al (2009), Xu et al (2009), van Dijk et al (2011), and Schulz et al (2013). CUT&RUN data were obtained GSE116853 (Brahma & Henikoff, 2019). Pol II CRAC data were described in GSE97913 and GSE97915 (Candelli et al,

2018). Reb1-ChEC-seq data were described in GSE67453 (Zentner et al, 2015).

# Data Availability

The accession number for the RNA sequencing data reported in this paper is GSE179256.

# Supplementary Information

# Acknowledgements

We thank the Crick Advanced Sequencing, Fermentation, Peptide Chemistry, and Genomics Equipment Park Facilities for experimental support, and Sebastian Marquardt and Jesper Svejstrup for sharing reagents. This research was funded in whole, or in part, by the Wellcome Trust (FC001203). For the purpose of Open Access, the author has applied a CC BY public copyright licence to any Author Accepted Manuscript version arising from this submission. This work was supported by the Francis Crick Institute (FC001203), which receives its core funding from Cancer Research UK (FC001203), the UK Medical Research Council (FC001203), and the Wellcome Trust (FC001203). We also thank the three reviewers for their constructive feedback throughout the review process.

## Author Contributions

ACK Wu: conceptualisation, data curation, formal analysis, validation, investigation, visualisation, methodology, writing—original draft, review, and editing, and the first and second authors contributed equally.
C Vivori: data curation, software, formal analysis, investigation, visualisation, methodology, writing—original draft, review, and editing, and the first and second authors contributed equally.
H Patel: data curation, software, formal analysis, validation, and methodology.
T Sideri: investigation and writing—review and editing.
F Moretto: formal analysis, investigation, and writing—review and editing.
FJ van Werven: conceptualisation, formal analysis, investigation, and writing—original draft, review, and editing.

## Conflict of Interest Statement

The authors declare that they have no conflict of interest.

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
