## [Reviewer comments · Life Science Alliance]

Life Science Alliance

RSC and GRFs confer promoter directionality by restricting divergent noncoding transcription

Andrew Wu, Claudia Vivori, Hashil Patel, Theodora Sideri, Fabien Moretto, and Folkert van Werven

DOI: <https://doi.org/10.26508/lsa.202201394>

Corresponding author(s): Folkert van Werven, The Francis Crick Institute

Review Timeline:

Submission Date:	2022-01-31
Editorial Decision:	2022-03-10
Revision Received:	2022-07-27
Editorial Decision:	2022-08-26
Revision Received:	2022-09-02
Accepted:	2022-09-02

Scientific Editor: Novella Guidi

Transaction Report:

March 10, 2022

Re: Life Science Alliance manuscript #LSA-2022-01394

Dr. Folkert van Werven
The Francis Crick Institute
1 Midland Road
London NW1 1AT
United Kingdom

Dear Dr. van Werven,

Thank you for submitting your manuscript entitled "RSC and GRFs confer promoter directionality by limiting divergent noncoding transcription" to Life Science Alliance. The manuscript was assessed by expert reviewers, whose comments are appended to this letter. We, thus, encourage you to submit a revised version of the manuscript back to LSA that responds to all the reviewers' points.

Thank you for this interesting contribution to Life Science Alliance. We are looking forward to receiving your revised manuscript.

Sincerely,

B. MANUSCRIPT ORGANIZATION AND FORMATTING:

Reviewer #1 (Comments to the Authors (Required)):

Wu et al present a manuscript entitled "RSC and GRFs confer promoter directionality by limiting divergent noncoding transcription". This work aims to examine promoter directionality in yeast and the nature of RSC and GRF effects on this property. The key issue with the work, which should be addressable rhetorically, is the conflation of mechanistic effects with observations that do not necessarily firmly support one or another mechanism. Throughout the manuscript there are descriptions of "repressing divergent transcription" when in many cases the differences in directionality are determined from relative comparisons. A clear discussion early in the manuscript of the possibilities for directionality - promoting one direction, inhibiting one direction, doing both - and discussion of possibilities that reduction in transcription in one direction could result in a relative change, or additionally, a shift in preference such that there might also be an accompanying increase in the divergent direction. This type of discussion early in the manuscript, coupled with conservatism in language when results are discussed could be clarifying. Throughout the manuscript it seems like different possibilities are usually treated under one umbrella model where divergent transcription is actively restricted by direct mechanisms. This appears to involve assumptions about mechanism that may not be supported by the nature of the experiments. Comments that may be useful to the authors are indicated below. One critical missing piece of data bookkeeping is evidence of reproducibility for any of the genomic data. One supplemental figure shows visual evidence (metagene, heat maps) for reproducibility of the Cut and Run, but no figures are generated for the RNA-seq data (consider heat maps, PCA, box plots, R2, etc). Preference would be demonstration reproducibility across features and metrics (directionality).

Title

1. "RSC and GRFs confer promoter directionality by limiting divergent noncoding transcription"

The title indicates mechanism that divergent transcription is limited- this suggests repression of divergent transcription. This conclusion would only be possible if measurements were absolute and not relative. It needs to be explicit, and reiterated, which measurements are comparable for levels vs only relative changes, and this should be repeatedly indicated as appropriate.

Abstract

2. "...noncoding transcription - is highly variable and regulated"

This could be semantics but maybe "regulated" is unclear and could imply that it is toggled at specific promoters when there might just be systems in place to promote directionality- which might suggest control/restriction but not necessarily regulation.

3. "At gene promoters that are highly directional, depletion of RSC leads to a relative increase in divergent noncoding transcription and thus a decrease in promoter directionality."

Here abstract is careful to indicate that RSC effect is relative increase but much language in paper seems to imply that it is absolute in some senses. This should be carefully addressed to make sure statements are appropriately conservative. Relative increase could be different than limitation of divergent transcription, so will want to be careful here depending on what the results actually are.

Introduction

4. It could be helpful to introduce concept that promoter regions are basically pairs of core promoters that drive divergent transcription from a shared NDR/NFR. This might help clearly define what is meant by promoter but also introduce concept of core promoter as divergent transcription is really the balance between these two core promoters at a promoter region. This then can allow concept of intrinsic property to be emphasized.

5. "Divergent and coding gene transcription share the same promoter sequence and the ratio..."

"Shared" promoter sequence here will not be clear unless introducing the concepts described above.

6. "Accordingly, promoters with a high ratio of divergent over coding transcription may have evolved fortuitously, and these promoters have been proposed to represent a transcriptional ground-state"

"those" is unclear here, as written appears to refer to promoters with high ratio of divergent over coding- I think what is meant here is promoters, upon fortuitous evolution, are proposed to be in a ground state of relatively balanced divergent transcription and subsequent evolution restricts one direction such that they might become increasingly directional. I think this will be more clear?

7. p5. "These promoters tend to be highly directional.."

Perhaps "the affected promoters" will be more clear here.

8. p5, bottom "Finally, we demonstrate that ectopic targeting of GRFs and dCas9 to divergent"

-> "GRFs or dCas9"

Results

9. Please indicate time of IAA treatment in main text and make clear on figure.

10. Figure 1. Examples in 1CD might be marked on scatter in 1B so reader can have idea of where these examples are in the distribution of promoters.

11. p7 "This decrease was lesser in nascent RNA-seq likely, in part, because internal normalization was used for the analysis."

This should be made more clear, as well as always indicating when effects are relative.

12. p8 "Notably, we observed increased RNA expression in the protein-coding direction in Rap1-depleted and Rap1/Sth1 codepleted cells (Figure S1C)."

It is hard to tell from these heat maps what effects are specific for the loci in question and what effects are reproducible. It would be nice to have some sort of presentation that made it crystal clear what is reproducible and that the effects are relative to each direction. The red on the heat map for one strand seems to just cover the entire region and this doesn't make a lot of sense.

13. p8 "These data suggest that RSC depletion caused a relatively greater in divergent transcription compared"

Is there a missing word here (after greater)?

14. p8 "While our analysis using nascent RNA-seq cannot determine absolute effects in transcription changes, it is possible to dissect relative changes."

It is essential that the language here is reiterated throughout manuscript and that descriptions of observed effects do not stray into more mechanistic language that is not supported.

15. p8 "Upon Rap1 depletion, directionality was strongly reduced due to both the increased in the divergent transcription and a decrease in sense transcription (Figures 2C and 2D)."

Just as example, when referring to any results please always indicate mRNA or nascent, and relative or not so that this will always be clear to reader.

16. p8 bottom of page "(Figures 2B and 2D)." Is this 2C and D?

17. p9 "Divergent noncoding transcription is an inherent feature of gene regulation across species"

What is meant by "regulation" is not clear here.

18. p10 "We divided gene promoters into quintiles (Q1-Q5) based on directionality score in WT cells, wherein Q5 represents the group of promoters with the highest directionality score and Q1 the lowest."

Do directionality quintiles correlate with expression level either mRNA or nascent? Consider also looking at directionality as scatter vs effects to see what amount of variation might be explained by directionality and to rule out binning effects.

19. p11 "Specifically, we compared profiles of MNase sensitivity in cells treated with high and low amounts of MNase using a published dataset"

Please consider here or in supplement showing heat maps separated into classes but rank ordered within classes by width of NFR region (distance between +1/-1) as determined by high MNase digestion. This will make it potentially much easier to look at where effects are or if there is any relationship between width of NFR/presence of fragile nucleosome and effects on different measured properties as the heat maps will likely be much more visually obvious. Also can consider looking at affected promoters for differences in attributes such as width of NFR (very likely can be used as proxy for presence of a fragile nucleosome). Can then show difference maps with this ordering as well.

20. p12 "gene promoters renders genome-wide changes"

"renders" appears to be the wrong word.

21. p12 "(Figure 4C, right panel), suggesting a prevalence for partially unwrapped nucleosomes in this group..."

Based on what? the larger difference between high and low? Might be more obvious in a heat map for each quintile ranked by apparent NFR width determined from high MNase- also could do box plots on widths- this relates to comment above on how to make the MNase data more accessible.

22. p14 "Interestingly, the A track on the antisense strand showed two peaks in promoters with highest the directionality, possibly indicating..."

Not clear if these are coming from same promoters and where they are relative to NDR widths, Could be worth displaying on heat map with rank ordering like proposed for MNase data.

23. p14 "Altogether, these data further support a model where RSC is recruited to distinct positions to promote positioning of upstream nucleosomes in promoters, thereby limiting divergent transcription initiation."

Because it is not clear what mechanism is- for example if there is some relationship between sense and divergent and if sense becomes more restricted than divergent, then the increased divergent is a consequence of decreased sense, limitation would only be indirect, or if effects really are only relative- depending on what is actually going on, there could be different models. If these ideas are made more clear it will be easier to consider the right context for the results.

24. p14 "The highly positioned nucleosomes, in turn, can form physical barriers which inhibit recruitment of the PIC and RNA Pol II."

This sentence is not clear and does not explicitly follow what comes before. Please clarify.

25, p15 "This suggests that, apart from Rap1, other GRFs may contribute to repressing divergent transcription."

This could be promotion of one direction, which would only be relative decrease in divergent.

26. p16 "Notably, Gal4 and Gcn4 showed increased signal in the coding direction, suggesting that they are still somewhat active in the conditions tested but perhaps not sufficiently active for repressing divergent transcription (Figure S6A)."

The two directions should be shown in the same figure as well as the calculated directionality. A model could be there is competition between the two directions and affecting one might also inversely affect other. It should be noted Gal4 will be bound along with Gal80 (therefore binding but not active as activator) in glucose and would need to be discussed in light of the dCas9 results. Also potentially just looking at DBD alone and asking if expression level of DBD (presumably scaling with occupancy) affects results? Experiments with activators instead of inert DBD are much harder to make conclusions from. This could be more compelling if just DBDs and not activators used, if both directions are examined in the main figure, and things that are not very big (like Cas9) were examined.

Discussion

27. p18 "Furthermore, we provided evidence that a wide range of DNA-binding proteins, including nucleosomes, GRFs, and dCas9, are capable of repressing divergent transcription."

What is the evidence that nucleosomes are repressing divergent transcription- is this just via RSC? The DNA binding proteins do not all appear to work the same way. A number appear to affect directionality but effects are mixed whether this is through one direction or the other.

28. p18 "DNA binding factors form physical barriers at promoters which prevent transcription in the divergent direction, thereby promoting promoter directionality."

It should be made explicit that, when there is direct evidence on absolute effects, that this model is supported when only one

direction (the divergent) is affected.

29. Figure 1E and related. It is not clear what is shown- is it just the one cluster out of 3 (for Rap1 affected genes) or if there are three clusters of promoters, how are they separated?

30. All figures in general: make it clear what n are for every figure, including whether western or northern blots are representative or are n=1.

31. Figure S2 How are all genes handled when not in tandem in terms of looking at same pair from both directions?

Cross comments: the other reviews are quite valuable and should aid in improving and revising the manuscript

Reviewer #2 (Comments to the Authors (Required)):

The ms by Wu et al., describes their finding on the role of chromatin remodelling complex components (RSC) and general regulatory factors (GRF) in regulating directionality of transcription from divergent promoters of coding and non-coding genes in *C. cerevisiae*. The authors use loss of function of RSC and GRF factors and subsequent genome wide analysis of nascent and polyA RNA produced at divergent promoters. They classify genes according to impact in the directionality of transcription measured by assessing the ratio of transcripts in both directions. They imply that the roles of RSC component Sth1 in nucleosome positioning at the divergent promoter region and argue for the role of GRFs Abf1 and Reb1 in inhibiting upstream transcription at directional promoters. The latter work which is only loosely linked to the first part of the ms is based on CUT and RUN-based binding analyses, loss of function and overexpression studies. Finally, they conclude that these factors facilitate regulation of directionality by steric hindrance on transcription machinery binding.

The evidence for the role of RSC factors in directionality is based on convincing LOF experiments. The subsequent analysis of the mechanism of the directionality regulation and analysis of GRF function are less robust, primarily based on binding profiling and to a smaller extent in manipulation of candidate binding sites that are ectopically inserted into a test system they created for reporter analysis of divergent promoters. The following queries arise from the presented work:

1. Abstract. RSC's role in nucleosome positioning is highlighted. As presented in Figure 4, the Sth1 depleted signal seems not to be substantially different from that in the wt. There is modest evidence for the difference being more pronounced within the Q5 set of promoters, and these observations do not make a very strong case for RSCs mediating directionality via nucleosome positioning.

2. Presentation of the results of nascent and mRNA seq in Fig. 1E could be improved by indicating the clustering (c1-C3) on the panels. I am confused by the labels in Fig S1C. The legend states that the analyses were matching that of Fig. 1E, yet the caption on the third panel is different.

In this analysis the data is aggregated to the Rap1 binding site. Since the authors argue about a steric effect by the RSC factor, it is conceivable that the effect on directionality is influenced by the positioning of the binding site in relation to either the downstream or upstream promoter. Therefore, it would be helpful to show heatmaps in which the genes are ranked by directionality index (and/or to normalised divergent transcription levels) and aligned to Sth1 sites. Similarly, promoters clustered in the Q1-Q5 could be ranked and the effect of potentially varying positioning of the Sth1 sites in relation to the TSS detected.

3. The presentation of the analysis of nucleosome positioning is somewhat confusing. In Fig. 4B,C fragile nucleosome occupancy is highlighted whereas panels C,D highlight positioning differences. Which of them is what the authors claim to be regulatory step, either or both? Judging from the presented data fragile nucleosome loss is similarly occurring between Q1 and Q5, it is the overall nucleosome occupancy within the promoter region what is more different in Q5 promoters (lower).

4. The polyA and CG(C/G)G motif analysis is intriguing, but the data presentation does not help in making clear conclusion on their relevance or robust association with candidate regulator binding sites. Firstly, in most panels the x axis scale varies and comparison between relevant and highly related datasets such as TBP binding, nucleosome positioning and directionality scores in relation to candidate RSC sites is not possible to make. For example, the authors argue that the two peaks seen in Fig.4E indicate multiple RSC localisations to influence nucleosome positioning. However, only the upstream peak appears to be specific to the Q5 group making the upstream peak a stronger candidate for regulatory role. If this polyA stretch is relevant to directionality associated RSC binding, it should align to the observed Sth1 binding sites in Fig 5 in Q5 promoters. It is currently not possible to assess. All of the positionally analysed data should be scaled in the various figures such, that detection of positional correlations is not obscured. BTW, Fig. 4C,D the notch indicating the TSS is hardly visible. This harmonisation should lead to better association between chromatin features and divergent TSS usage determinants.

5. The periodicity of polyA in Fig S4 is intriguing. What could be the reason for this? Is it due to nucleosome positioning AT/GC frequency?

6. The analysis of TBP binding upon Sth1 loss is again hard to interpret as it is not clear how far the TBP bound TATA boxes and TATA like elements are in relation to the dominant TSS (expected at about -30bp).

7. The design of the manipulation experiments presented in Fig. 6 is not clearly justified. Based on the aggregated factor binding profiles and various TSS distance relationship analyses it is difficult to appreciate why the authors chose positioning their manipulated binding sites 20bp away from the divergent transcript. This design is likely based on the Sco1 example (hard to judge from the image without a scale bar), however, this example seems not to fit the majority positioning. Aggregated Cut and Run results presented in Fig. 5 suggest that Abf1 and Reb1 peaks with most dramatic impact on directionality suggest

preference in relation to the downstream TSS. Is there a positional preference in relation to the upstream TSS? The reporter assays should allow testing such positioning preferences in relation to both TSS.

Minor points:

I suggest that the authors revise the formatting of their manuscript such that the figure legends are printed on the same page under the relevant panels. This massively improves the readability. The figures could be squeezed and the legend printed at smaller font in single space to facilitate more convenient reading.

P5 bottom para, 'enriched for GRFs' should be 'enriched for GRF binding' or similar.
In the sentence before, delete 'in'

P7 'promoters more closely' - do the authors mean 'more broadly' or 'across Rap1 regulated genes'?

P8 top para: greater.. (a word seems to be missing here)

Genome browsers shots need x axis size indicator.

Fig. 2A size bar is missing

Fig. 5A. There is a word 'signal' in the third panel. What does this exemplify?

Reviewer #3 (Comments to the Authors (Required)):

This paper describes an in-depth study of the mechanisms underlying divergent non-coding transcription. It is unclear how this is regulated. The authors show that the RSC complex and general regulatory factors (GRFs) contribute to repression of divergent transcription in wild type cells. Depletion of Sth1, the ATPase subunit of the RSC complex, reduces transcription of genes regulated by the Rap1 GRF; other GRFs have similar effects. The authors performed nascent RNA-seq and standard RNA-seq on wild type yeast cells and cells depleted of Sth1 or Rap1, or both. They propose that RSC-mediated nucleosome positioning over the divergent promoter is responsible for repression of divergent transcription. This is an interesting study of an important question. I have the following comments:

1. Depletion of Sth1 reduces transcription of Rap1-regulated genes by ~2-fold as measured by mRNA-seq, but there is little or no change when measured by nascent RNA-seq (Fig. 1B). This discrepancy is not discussed properly. On p.7, it is stated that, in Sth1-depleted cells, the decrease of nascent RNA is less than that of mRNA, but nascent RNA actually shows no effect (Fig. 1B left panel). Wild type control data are missing from Fig. 1B. This is the main concern - why are these two data sets giving opposite results (see #2 and #5 below also)?
2. Overall, Sth1 depletion has a mild effect on anti-sense strand transcription, as measured by nascent RNA-seq (Fig. 1E, panel 1), but the effect as measured by mRNA-seq seems marginal (Fig. 1E, panel 4). Again, there is a discrepancy between the two measurements. Since this Rap1-independent effect of RSC is a key conclusion of the paper, the problem needs to be resolved. Wild-type control data should be added to Fig. 1E.
3. What is the explanation for the increase in anti-sense transcription over the Rap1-regulated gene ORFs after Rap1 depletion (Fig. 1E, panel 5, righthand side) and, even more so, after depletion of Rap1 and Sth1 (Fig. 1E, panel 6, righthand side)?
4. The density plots in Figure 2 show peaks corresponding to about half of the number of genes plotted, which seems problematic given the area under the curve. Have the data been smoothed too much? They should really be presented as a histogram. The effect of Sth1 depletion (Fig. 2E) is small - is it just an artefact of over-smoothing? This matters because these data underpin the conclusion that RSC represses divergent transcription.
5. Fig. 2C shows a plot comparing fold-changes in nascent RNA-seq for the upstream RNA and the sense RNA. The effect is quite clear, but is it confirmed by the mRNA-seq data?
6. On page 12, the effect of Sth1 depletion on promoter chromatin structure is addressed (Fig. 4B,C and Fig. S4). The fragile nucleosome/MNase-sensitive complex peak observed at low MNase digestion in the PDE2 promoter (Fig. 4B, right panel) seems to be a rare exception, since there is no such peak in the average plot (Fig. 4C, righthand panel). This fact should be pointed out. Also, it is stated that "Third, the +1, -1 and -2 nucleosomes were more sensitive to MNase concentration in Q5 promoters than Q1 promoters (Figure 4C, right panel), suggesting a prevalence for partially unwrapped nucleosomes in this group." This conclusion is dubious. Since linkers are digested at different rates depending on their sequence, the relative nucleosome peak intensity at low levels of MNase may not be quantitatively reliable. Is this subtle effect reproducible? Analysis of the replicate experiments is needed. A lower nucleosome peak does not necessarily indicate fewer nucleosomes; weaker phasing flattens the peaks and fills the troughs (the extreme case is random positioning, which predicts a flat line at the average

value). An alternative interpretation might be that the nucleosome-depleted region (NDR) at these promoters is widened in some Sth1-depleted cells, as the divergent promoter is activated for transcription. The assumption that fragile nucleosomes are involved also appears in the Discussion (p. 18), without the appropriate caveat (no definitive evidence is presented here for fragile nucleosomes on divergent promoters).

7. Fig. 4D: Sth1 depletion reduces TBP levels at the promoters of genes in quintile 5. Is this small effect reproducible? Analysis of the replicate experiments is needed.

8. Discussion, page 18. It is stated that "Here, we showed that RSC mediated nucleosome positioning represses divergent noncoding transcription." What is shown is only that there is a correlation between RSC-mediated nucleosome positioning and repression of divergent noncoding transcription; cause and effect have not been distinguished.

9. A more general issue is that the cells are dying as essential proteins (Sth1 and/or Rap1) are depleted, perhaps activating promoters that are never activated in wild type cells. There is evidence that RSC depletion results in a general decrease in transcription (Parnell et al. 2008, cited). How might this affect the measurement of divergent transcription? The authors could comment on this in the Discussion.

Minor points:

10. I found the term "divergent transcription" confusing, because divergent transcription is often used to mean transcription of two genes from a central promoter, implying that transcription occurs in both directions. Here, divergent transcription refers to transcription of the gene upstream of the Rap1-regulated gene (i.e. transcription of the upstream gene only). For example, in the section title "RSC represses divergent transcription independently of Rap1", the authors are presumably referring only to the upstream gene, not both genes (since transcription of the downstream, Rap1-regulated, gene is activated by RSC, not repressed; Fig. 1B). However, I don't know how to fix the terminology!

11. The IAA concentration used is missing.

12. Fig. 1A: some bands seem to have been cut off in the Western.

13. The double-depletion data (Sth1 and Rap1) are missing from Fig. 1D.

Reviewer #1 (Comments to the Authors (Required)):

Wu et al present a manuscript entitled "RSC and GRFs confer promoter directionality by limiting divergent noncoding transcription". This work aims to examine promoter directionality in yeast and the nature of RSC and GRF effects on this property. The key issue with the work, which should be addressable rhetorically, is the conflation of mechanistic effects with observations that do not necessarily firmly support one or another mechanism. Throughout the manuscript there are descriptions of "repressing divergent transcription" when in many cases the differences in directionality are determined from relative comparisons. A clear discussion early in the manuscript of the possibilities for directionality - promoting one direction, inhibiting one direction, doing both - and discussion of possibilities that reduction in transcription in one direction could result in a relative change, or additionally, a shift in preference such that there might also be an accompanying increase in the divergent direction. This type of discussion early in the manuscript, coupled with conservatism in language when results are discussed could be clarifying. Throughout the manuscript it seems like different possibilities are usually treated under one umbrella model where divergent transcription is actively restricted by direct mechanisms. This appears to involve assumptions about mechanism that may not be supported by the nature of the experiments. Comments that may be useful to the authors are indicated below. One critical missing piece of data bookkeeping is evidence of reproducibility for any of the genomic data. One supplemental figure shows visual evidence (metagene, heat maps) for reproducibility of the Cut and Run, but no figures are generated for the RNA-seq data (consider heat maps, PCA, box plots, R2, etc). Preference would be demonstration reproducibility across features and metrics (directionality).

We thank the reviewer for the comments. As detailed below, we have updated the manuscript. We think that the revised manuscript clarified the effects on promoter directionality by RSC and GRFs.

We have now included PCA plots of the nascent-RNA-seq and mRNA-seq data in Figure S1B. We performed all genome-wide experiments using 3 biological replicates. All biological replicates clustered well together, as can be seen in the PCA plots.

1. "RSC and GRFs confer promoter directionality by limiting divergent noncoding transcription"

The title indicates mechanism that divergent transcription is limited- this suggests repression of divergent transcription. This conclusion would only be possible if measurements were absolute and not relative. It needs to be explicit, and reiterated, which measurements are comparable for levels vs only relative changes, and this should be repeatedly indicated as appropriate.

We thank the reviewer for the comment. We agree that nascent transcription RNA-seq is a relative measurement. We have indicated, where possible, when we described the relative measurements. Noteworthy, the mRNA-seq can be considered an absolute measurement as it is normalized over *S. Pombe* RNA and it does show a similar (although less strong) effect as the nascent RNA-seq. We intended to change the title to:

"RSC and GRFs confer promoter directionality by restricting relative levels of divergent noncoding transcription"

However, LSA has a 100-character title limit, hence have decided to opt for the following title:

"RSC and GRFs confer promoter directionality by restricting divergent noncoding transcription"

2. "...noncoding transcription - is highly variable and regulated"

This could be semantics but maybe "regulated" is unclear and could imply that it is toggled at specific promoters when there might just be systems in place to promote directionality- which might suggest control/restriction but not necessarily regulation.

We have removed "regulated" in line 2, as we agree it could be misleading. Indeed, we do not know whether there is deliberate regulation of divergent transcription.

3. "At gene promoters that are highly directional, depletion of RSC leads to a relative increase in divergent noncoding transcription and thus a decrease in promoter directionality."

Here abstract is careful to indicate that RSC effect is relative increase but much language in paper seems to imply that it is absolute in some senses. This should be carefully addressed to make sure statements are appropriately conservative. Relative increase could be different than limitation of divergent transcription, so will want to be careful here depending on what the results actually are.

We thank the reviewer for the comment. We agree. We went over the manuscript and now have changed the language describing when relative measurements were made.

Introduction

4. It could be helpful to introduce concept that promoter regions are basically pairs of core promoters that drive divergent transcription from a shared NDR/NFR. This might help clearly define what is meant by promoter but also introduce concept of core promoter as divergent transcription is really the balance between these two core promoters at a promoter region. This then can allow concept of intrinsic property to be emphasized.

We have added one sentence introducing the concept suggested by the reviewer. Line 46:

"Within promoters, the divergent and coding core promoters are pairs of core promoters which typically share the same nucleosome depleted region (NDR) where transcription initiates in divergent directions."

5. "Divergent and coding gene transcription share the same promoter sequence and the ratio..."

"Shared" promoter sequence here will not be clear unless introducing the concepts described above.

We have removed the part on shared promoters (line 49).

6. "Accordingly, promoters with a high ratio of divergent over coding transcription may have evolved fortuitously, and these promoters have been proposed to represent a transcriptional ground-state"

"those" is unclear here, as written appears to refer to promoters with high ratio of divergent over coding- I think what is meant here is promoters, upon fortuitous evolution, are proposed to be in a ground state of relatively balanced divergent transcription and subsequent evolution restricts one direction such that they might become increasingly directional. I think this will be more clear?

We agree and changed the section accordingly (line 56):

"Promoters, upon fortuitous evolution, are proposed to be in a ground state of relatively balanced divergent transcription and subsequent evolution restricts one direction such that they might become

increasingly directional (Jin et al., 2017). As such, evolved promoters are proposed to be highly directional, thus displaying little divergent transcription but high levels of sense coding gene transcription."

7. p5. "These promoters tend to be highly directional.."
Perhaps "the affected promoters" will be more clear here.

We agree. We changed this accordingly (line 104).

8. p5, bottom "Finally, we demonstrate that ectopic targeting of GRFs and dCas9 to divergent"
-> "GRFs or dCas9"

The text was changed accordingly (line 109).

Results

9. Please indicate time of IAA treatment in main text and make clear on figure.

The cells were treated with IAA for 2 hours to induce the auxin-inducible degron. This is now indicated in the main text and figure (line 125).

10. Figure 1. Examples in 1CD might be marked on scatter in 1B so reader can have idea of where these examples are in the distribution of promoters.

We have now highlighted RPL43A and RPS10A in Figure 1B.

11. p7 "This decrease was lesser in nascent RNA-seq likely, in part, because internal normalization was used for the analysis."

This should be made more clear, as well as always indicating when effects are relative.

We went through the manuscript and have indicated where observations are relative.

12. p8 "Notably, we observed increased RNA expression in the protein-coding direction in Rap1-depleted and Rap1/Sth1 codepleted cells (Figure S1C)."

It is hard to tell from these heat maps what effects are specific for the loci in question and what effects are reproducible. It would be nice to have some sort of presentation that made it crystal clear what is reproducible and that the effects are relative to each direction. The red on the heat map for one strand seems to just cover the entire region and this doesn't make a lot of sense.

We thank the reviewer for the suggestion. The data are highly reproducible, as shown by the PCA in Figure S1B. We have removed the heatmap in S1C, because it presents aberrant transcription in the sense or 'coding' direction around the Rap1 binding sites, which is not directly relevant for this study. The reason why we observed a significant increase in the Sth1 and Rap1 double depletion is because Sth1 is also known to repress antisense transcription. Thus, likely upon Rap1 depletion more antisense transcription is induced when Sth1 is also depleted. However, this observation is less relevant to the message of the manuscript. We clarified the reproducibility and use of replicates in the text and figure legends.

As suggested by reviewer 3, we have now included data of the DMSO/WT comparison for the antisense direction. We agree that the heatmaps do not clearly quantify differences and are more useful in identifying the genomic locations where changes in RNA expression occur. Therefore, we

quantified the signal on the divergent and sense strands in Figure 2C which shows the quantification of Rap1-regulated genes. This graph also includes the corresponding of examples in Figure 1C. Moreover, we repeated the quantification of divergent and sense signals performed in Figure 2C on the mRNA-seq data, which is shown in Figure S2B. Both graphs show clearly that, on average, divergent transcription is upregulated and sense transcription is down regulated.

13. p8 "These data suggest that RSC depletion caused a relatively greater in divergent transcription compared"

Is there a missing word here (after greater)?

Indeed, it should read "greater increase".

14. p8 "While our analysis using nascent RNA-seq cannot determine absolute effects in transcription changes, it is possible to dissect relative changes."

It is essential that the language here is reiterated throughout manuscript and that descriptions of observed effects do not stray into more mechanistic language that is not supported.

We have clarified this and indicated where findings are relative throughout the manuscript.

15. p8 "Upon Rap1 depletion, directionality was strongly reduced due to both the increased in the divergent transcription and a decrease in sense transcription (Figures 2C and 2D)."

Just as example, when referring to any results please always indicate mRNA or nascent, and relative or not so that this will always be clear to reader.

We have clarified this and indicated where findings are relative throughout the manuscript. We changed the sentence to:

"Upon Rap1 depletion, directionality as determined by nascent RNA-seq was strongly reduced due to increased divergent transcription and decreased in sense transcription (Figures 2C and 2D)."

16. p8 bottom of page "(Figures 2B and 2D)." Is this 2C and D?

It is 2C and 2D. We have corrected this now.

17. p9 "Divergent noncoding transcription is an inherent feature of gene regulation across species"

What is meant by "regulation" is not clear here.

Indeed, regulation is not the accurate term. We have rephrased this into the following (line 235):

"Divergent noncoding transcription is an inherent feature of active promoters across eukaryotic species"

18. p10 "We divided gene promoters into quintiles (Q1-Q5) based on directionality score in WT cells, wherein Q5 represents the group of promoters with the highest directionality score and Q1 the lowest."

Do directionality quintiles correlate with expression level either mRNA or nascent? Consider also looking at directionality as scatter vs effects to see what amount of variation might be explained by directionality and to rule out binning effects.

We thank the reviewer for the comment. We have generated the scatter plots in S4E and S4F. In the plot that is sorted on directionality, we also observe a positive correlation between directionality score and increased divergent transcription upon Sth1 depletion. When we sort according sense

transcription levels, we see less of an effect on divergent transcription when Sth1 is depleted, compared to when we sort for directionality. We acknowledge that the directionality and sense transcription levels are linked. Indeed roughly 50% of genes in Q5 of the directionality and sense transcription overlap. Nevertheless, our data suggest that RSC represses divergent at promoters with high directionality score, which may be partially linked the sense transcription levels.

19. p11 "Specifically, we compared profiles of MNase sensitivity in cells treated with high and low amounts of MNase using a published dataset"

Please consider here or in supplement showing heatmaps separated into classes but rank ordered within classes by width of NFR region (distance between +1/-1) as determined by high MNase digestion. This will make it potentially much easier to look at where effects are or if there is any relationship between width of NFR/presence of fragile nucleosome and effects on different measured properties as the heat maps will likely be much more visually obvious. Also can consider looking at affected promoters for differences in attributes such as width of NFR (very likely can be used as proxy for presence of a fragile nucleosome). Can then show difference maps with this ordering as well.

We thank the reviewer for her/his comment. We have generated metagene plots and heatmaps that outline the differences in nucleosome occupancy based on the date form (Kubik et al., 2015, Molecular Cell). From the heatmaps in Figure 4D is it clear that fragile nucleosomes are enriched in Q5 (see also figure S4G). We also have analysed a second dataset that measured nucleosome occupancy upon Sth1 depletion in 2 biological replicates (Figure S4H). {Klein-Brill, 2019 #185. Also, in this dataset we observe that -1/FN is better positioned in Q5 vs Q1.

Noteworthy, the NDR annotation was missing for many genes, making the representation of quintiles uneven (for example, only 289 genes out of 522 of Q1 have correct NDR length data, while 499 out of 521 in Q5 are represented). Hence, we have not included the data sorted on NDR length but have included these here below. In our view, the data as presented in the paper show clear differences in the -1/FN nucleosome signal. Sorting the data on NDR length does not make much difference.

Moreover, we have quantified the number of fragile nucleosomes for each class (Figure S4G) and observe a correlation between directionality and the number fragile nucleosomes. Lastly, we also assessed whether NDR width correlates with directionality as suggested. The violin plots below show there is no obvious trend, indicating the NDR width is perhaps not a determinant for promoter directionality (Q1 = 102-157nt, Q2 = 158-190nt, Q3 = 191-238nt, Q4 = 239-266nt, Q5 = 267-473nt). We have not included this in the current manuscript.

20. p12 "gene promoters renders genome-wide changes"
 "renders" appears to be the wrong word.

We changed this to “reflect” (line 326).

21. p12 "(Figure 4C, right panel), suggesting a prevalence for partially unwrapped nucleosomes in this group..."

Based on what? the larger difference between high and low? Might be more obvious in a heat map for each quintile ranked by apparent NFR width determined from high MNase- also could do box plots on widths- this relates to comment above on how to make the MNase data more accessible.

We have now quantified the number of fragile nucleosomes as defined in Figure S4G (Kubik et al., 2015, Molecular Cell). We observe a correlation between promoter directionality and the number of fragile nucleosomes present. This further supports the idea that fragile nucleosomes are more present at promoters with high directionality.

22. p14 "Interestingly, the A track on the antisense strand showed two peaks in promoters with highest the directionality, possibly indicating..."

Not clear if these are coming from same promoters and where they are relative to NDR widths, Could be worth displaying on heat map with rank ordering like proposed for MNase data.

We have now displayed the MNase-seq data for Q1 and Q5, including heatmaps (Figure 4D). We have also overlaid the A-track motif with RSC binding (Figure 5B). We find that the enrichment of A-tracks (sense and antisense) overlapped second peaks of RSC binding as determined by CUT and RUN. This suggests that A-tracks possibly contribute to recruiting RSC and thereby control directionality promoters. The analysis to determine where the A-tracks are with respect to NDR is not trivial for many promoters. First the A-track motif is rather degenerate, and thus many promoters have more than one A-track. Second, the NDR annotation is not complete, and we will therefore inevitably miss substantial fraction of promoters for the analysis.

23. p14 "Altogether, these data further support a model where RSC is recruited to distinct positions to promote positioning of upstream nucleosomes in promoters, thereby limiting divergent transcription initiation."

Because it is not clear what mechanism is- for example if there is some relationship between sense and divergent and if sense becomes more restricted than divergent, then the increased divergent is a consequence of decreased sense, limitation would only be indirect, or if effects really are only relative- depending on what is actually going on, there could be different models. If these ideas are made more clear it will be easier to consider the right context for the results.

We thank the reviewer for the comment. We agree alternative models need to be considered. Therefore, we have removed the statement from the results and have now included a section in the discussion (“RSC, nucleosome positioning and divergent transcription”) describing the different models (e.g. direct vs indirect).

24. p14 "The highly positioned nucleosomes, in turn, can form physical barriers which inhibit recruitment of the PIC and RNA Pol II."

This sentence is not clear and does not explicitly follow what comes before. Please clarify.

We thank the reviewer for the comment. We have now changed the sentence and moved this section to the discussion where we also describe other models.

" Alternatively, it is possible that RSC indirectly represses divergent transcription by stimulating coding sense transcription. Indeed, RSC also acts in coding regions to promote Pol II elongation (Biernat et al., 2021). Why RSC does not act at noncoding divergent regions is not clear. We favour a model where nucleosomes positioned by RSC form barriers to limit PIC formation and Pol II recruitment for the divergent direction (Figure 6J). Upon RSC depletion, the positioning of the -1 nucleosomes is less strong and PIC formation can occur."

25, p15 "This suggests that, apart from Rap1, other GRFs may contribute to repressing divergent transcription."

This could be promotion of one direction, which would only be relative decrease in divergent.

We agree that there are multiple models possible. Hence, we were cautious to state "suggest" and "may". We think it is not productive to discuss other models at this stage. We changed the sentence accordingly.

"This suggests that, apart from Rap1, other GRFs, in addition to promoting transcription in the coding direction, may contribute to repressing divergent transcription."

26. p16 "Notably, Gal4 and Gcn4 showed increased signal in the coding direction, suggesting that they are still somewhat active in the conditions tested but perhaps not sufficiently active for repressing divergent transcription (Figure S6A)."

The two directions should be shown in the same figure as well as the calculated directionality. A model could be there is competition between the two directions and affecting one might also inversely affect other. It should be noted Gal4 will be bound along with Gal80 (therefore binding but not active as activator) in glucose and would need to be discussed in light of the dCas9 results. Also potentially just looking at DBD alone and asking if expression level of DBD (presumably scaling with occupancy) affects results? Experiments with activators instead of inert DBD are much harder to make conclusions from. This could be more compelling if just DBDs and not activators used, if both directions are examined in the main figure, and things that are not very big (like Cas9) were examined.

We thank the reviewer for the comment. We have now included both coding signal as well as directionality calculations. We agree that a competition between two directions is possible. However, the signal in the coding direction does not correlate with the presence of transcription factors and their binding sites. For example, Gcr1, Cbf1, Abf1, and Reb1 show increased signal in the divergent noncoding direction when the transcription factors are depleted or deleted, while only coding direction signal decreased for the Cbf1 deletion mutant. Additionally, depletion/deletion of Gcn4 and Gal4 results in a decrease in the coding signal, but has no effect on the signal in the noncoding direction. We have commented on the constitutive binding of Gal4 in the results section.

Regarding the DNA binding domain experiment, the Libri lab has performed the experiment with Rap1 DNA-binding domain and that was able to repress aberrant transcription initiation at Rap1 regulated gene promoters (Challal et al., 2018, Molecular Cell), although the exact conditions under which this experiment was performed were not confirmed (*e.g.* expression level of the DNA binding domain and ability to bind DNA as measured by CHIP). We have commented on this in the discussion.

Discussion

27. p18 "Furthermore, we provided evidence that a wide range of DNA-binding proteins, including nucleosomes, GRFs, and dCas9, are capable of repressing divergent transcription."

What is the evidence that nucleosomes are repressing divergent transcription- is this just via RSC? The DNA binding proteins do not all appear to work the same way. A number appear to affect directionality but effects are mixed whether this is through one direction or the other.

We agree that we have not shown all that is stated in this sentence. The RSC data was already described in the previous sentence. Based on the reporter assay and northern blot, we are confident that Reb1 and Abf1 repress divergent transcription. We rephrased the sentence accordingly (line 460): "Furthermore, we provided evidence that other GRFs (Reb1 and Abf1) and dCas9 are capable of repressing divergent transcription."

28. p18 "DNA binding factors form physical barriers at promoters which prevent transcription in the divergent direction, thereby promoting promoter directionality."
It should be made explicit that, when there is direct evidence on absolute effects, that this model is supported when only one direction (the divergent) is affected.

We thank the reviewer for the comment. We have now rephrased the statement and emphasized the relative nature of the study by stating "restricts divergent noncoding transcription relative to coding gene transcription." Noteworthy, "we propose" leaves some room for interpretation. The statement (line 461) reads as follows:

"We propose that RSC-positioned nucleosomes and other DNA-binding factors form physical barriers in promoters which, in turn, restricts divergent noncoding transcription relative to coding gene transcription."

29. Figure 1E and related. It is not clear what is shown- is it just the one cluster out of 3 (for Rap1 affected genes) or if there are three clusters of promoters, how are they separated?

We have now included the clusters in Figure 1E. These were based on our previous work on Rap1 (Wu et al., 2018, Molecular Cell). The k-means clustering was performed on the antisense signal: clusters 1 and 2 display a large increase in divergent transcription, while in cluster 3 there was little divergent transcription in the Rap1 depletion.

30. All figures in general: make it clear what n are for every figure, including whether western or northern blots are representative or are n=1.

We have specified in each figure legends how many biological repeats were performed and represented.

31. Figure S2 How are all genes handled when not in tandem in terms of looking at same pair from both directions?

For the S2 figure non tandem promoters will be counted from both directions, thus from point of view of each gene. Since we have not done any further analysis on non-tandem gene promoters, we think the analysis is still valid.

Cross comments: the other reviews are quite valuable and should aid in improving and revising the manuscript

Reviewer #2 (Comments to the Authors (Required)):

The ms by Wu et al., describes their finding on the role of chromatin remodelling complex components (RSC) and general regulatory factors (GRF) in regulating directionality of transcription from divergent promoters of coding and non-coding genes in *C. cerevisiae*. The authors use loss of function of RSC and GRF factors and subsequent genome wide analysis of nascent and polyA RNA produced at divergent promoters. They classify genes according to impact in the directionality of transcription measured by assessing the ratio of transcripts in both directions. They imply that the roles of RSC component Sth1 in nucleosome positioning at the divergent promoter region and argue for the role of GRFs Abf1 and Reb1 in inhibiting upstream transcription at directional promoters. The latter work which is only loosely linked to the first part of the ms is based on CUT and RUN-based binding analyses, loss of function and overexpression studies. Finally, they conclude that these factors facilitate regulation of directionality by steric hindrance on transcription machinery binding.

The evidence for the role of RSC factors in directionality is based on convincing LOF experiments. The subsequent analysis of the mechanism of the directionality regulation and analysis of GRF function are less robust, primarily based on binding profiling and to a smaller extent in manipulation of candidate binding sites that are ectopically inserted into a test system they created for reporter analysis of divergent promoters. The following queries arise from the presented work:

1. Abstract. RSC's role in nucleosome positioning is highlighted. As presented in Figure 4, the Sth1 depleted signal seems not to be substantially different from that in the wt. There is modest evidence for the difference being more pronounced within the Q5 set of promoters, and these observations do not make a very strong case for RSCs mediating directionality via nucleosome positioning.

We thank the reviewer for the comment. We agree that we presented correlative data, and the effects are modest. Given that there is some effect on nucleosome positioning, and RSC enzymatic activity primary functions in positioning of nucleosomes, we think it is important to document this. It is also worth noting that the effects on divergent transcription are clear, but they are not large. Hence, we do not expect to observe large changes in nucleosome positioning. We have now included new analysis of a second dataset where we observe the same trend (Figure S4H). Accordingly, we have modified the statement in the abstract (line 9):

“We find that RSC has a modest effect on nucleosome positioning upstream in promoters at the sites of divergent transcription.”

2. Presentation of the results of nascent and mRNA seq in Fig. 1E could be improved by indicating the clustering (c1-C3) on the panels. I am confused by the labels in Fig S1C. The legend states that the analyses were matching that of Fig. 1E, yet the caption on the third panel is different.

In this analysis the data is aggregated to the Rap1 binding site. Since the authors argue about a steric effect by the RSC factor, it is conceivable that the effect on directionality is influenced by the positioning of the binding site in relation to either the downstream or upstream promoter. Therefore, it would be helpful to show heatmaps in which the genes are ranked by directionality index (and/or to normalised divergent transcription levels) and aligned to Sth1 sites. Similarly, promoters clustered in the Q1-Q5 could be ranked and the effect of potentially varying positioning of the Sth1 sites in relation to the TSS detected.

We apologise for the confusion and corrected the figures accordingly. We have indicated the clusters in Figure 1E. Figure S1C contained an error in the labelling: the same clusters of Figure 1E are presented, but with the signal on the sense strand (while the main figure shows the antisense strand). Noteworthy, we observed sense levels are increased upon Sth1 and Rap1 depletion which is consistent with previous reports (Challal et al., 2018). We have removed this panel as it did not add to the

message of the figure. We have now included a heatmap of antisense signal (DMSO vs. WT) to show that there is not much difference between mock-treated AID tag strains and WT (Figure S1E).

We agree that it would be ideal to dissect the contribution from motifs to RSC binding to RSC activity. However, this is not so straightforward because of the following reasons. Due to the degenerate nature of these sequences, it is hard to pick one motif to centre our analysis on based solely on the motif, as many promoters contain multiple A-tracks and GC-rich motifs. Typically, RSC associates with NDRs and can use A-track or GC-tracks as recognition sites. In general, RSC binding sequences remain not well defined because RSC is difficult to capture on DNA by chromatin IP. We have now overlaid the A-track and GC-rich motifs with CUT&RUN RSC binding (Figures 5B and S5B). We find that the enrichment of A-tracks (sense and antisense) overlapped with the second peaks of RSC binding as determined by CUT and RUN. This suggests that A-tracks could be responsible for bringing more RSC to directional promoters. A causal relationship between A-tracks and promoter directionality remains to be established.

3. The presentation of the analysis of nucleosome positioning is somewhat confusing. In Fig. 4B,C fragile nucleosome occupancy is highlighted whereas panels C,D highlight positioning differences. Which of them is what the authors claim to be regulatory step, either or both? Judging from the presented data fragile nucleosome loss is similarly occurring between Q1 and Q5, it is the overall nucleosome occupancy within the promoter region what is more different in Q5 promoters (lower).

We thank the reviewer for the comment. To keep things consistent, we labelled the FN to -1/FN as this likely presents a population of both fragile and stable -1 nucleosomes. Additionally, we have included a heatmap of the nucleosome positioning (Figure 4D and S4H).

We cannot conclusively say what the regulatory step is, but we make two observations. First, we observe that nucleosome positioning, which we define by the width of the peak at -2 and -1/FN, is more defined in Q5 vs Q1. Additionally, we observe the increased presence of fragile nucleosomes in Q5 vs Q1, which we now have quantified based on the annotation of (Kubik et al., 2015) and graphed in Figure S4G. Based on the Rap1-regulated genes described in Figure 1, we observe that divergent transcription is increased upstream of the Rap1 binding site, which is proximal to the -2 nucleosomes and more distal from the -1/FN nucleosomes (Kubik et al., 2018). So, in addition to the positioning of the upstream nucleosomes which is directly important, likely positioning of the -1/FN could influence the positioning of the upstream nucleosome, therefore affecting directionality.

Previous studies have shown that RSC positioned nucleosomes prevents aberrant transcription in the sense direction (Klein-Brill et al., 2019, Kubik et al., 2019). Also in these studies, the effect on nucleosome positioning were relatively marginal.

4. The polyA and CG(C/G)G motif analysis is intriguing, but the data presentation does not help in making clear conclusion on their relevance or robust association with candidate regulator binding sites. Firstly, in most panels the x axis scale varies and comparison between relevant and highly related datasets such as TBP binding, nucleosome positioning and directionality scores in relation to candidate RSC sites is not possible to make. For example, the authors argue that the two peaks seen in Fig.4E indicate multiple RSC localisations to influence nucleosome positioning. However, only the upstream peak appears to be specific to the Q5 group making the upstream peak a stronger candidate for regulatory role. If this polyA stretch is relevant to directionality associated RSC binding, it should align to the observed Sth1 binding sites in Fig 5 in Q5 promoters. It is currently not possible to assess. All of the positionally analysed data should be scaled in the various figures such that detection of positional correlations is not obscured. BTW, Fig. 4C,D the notch indicating the TSS is hardly visible. This

harmonisation should lead to better association between chromatin features and divergent TSS usage determinants.

We thank the reviewer for the comment. We have scaled the graphs accordingly. We also now include graphs that overlay the RSC motifs with RSC binding (Figures 5B and S5B).

5. The periodicity of polyA in Fig S4 is intriguing. What could be the reason for this? Is it due to nucleosome positioning AT/GC frequency?

We agree that periodicity is intriguing. Also, the fact that RSC binding motifs show some enrichment in directional promoters further supports the idea that RSC acts at directional promoters.

6. The analysis of TBP binding upon STh1 loss is again hard to interpret as it is not clear how far the TBP bound TATA boxes and TATA like elements are in relation to the dominant TSS (expected at about -30bp).

We acknowledge that this dataset does not have the resolution. The 3rd reviewer requested biological repeats of this experiment. Given that the published the datafile did not contain information on biological repeats, we decided to remove this experiment from the manuscript.

7. The design of the manipulation experiments presented in Fig. 6 is not clearly justified. Based on the aggregated factor binding profiles and various TSS distance relationship analyses it is difficult to appreciate why the authors chose positioning their manipulated binding sites 20bp away from the divergent transcript. This design is likely based on the Sco1 example (hard to judge from the image without a scale bar), however, this example seems not to fit the majority positioning. Aggregated Cut and Run results presented in Fig. 5 suggest that Abf1 and Reb1 peaks with most dramatic impact on directionality suggest preference in relation to the downstream TSS. Is there a positional preference in relation to the upstream TSS? The reporter assays should allow testing such positioning preferences in relation to both TSS.

We thank the reviewer for the comment. We based the 20 bp on our previous work with the transcription factor Rap1 (Wu et al., 2018). In this work, we found that divergent TSSs newly formed upon Rap1 depletion within 50 bp of the Rap1 binding site. Hence 20 bp, which is also the site of unique restriction site, was inside this range, close enough (based the Rap1 analysis) to interfere with divergent transcription. We have now included a paragraph describing the motivation for this (line 388):

“Previously we showed that Rap1 represses initiation of divergent transcription near its binding site, typically within 50 base pairs (bp) of its binding site (Wu et al., 2018).”

Minor points:

I suggest that the authors revise the formatting of their manuscript such that the figure legends are printed on the same page under the relevant panels. This massively improves the readability. The figures could be squeezed and the legend printed at smaller font in single space to facilitate more convenient reading.

We have done this for the revised manuscript.

P5 bottom para, 'enriched for GRFs' should be 'enriched for GRF binding' or similar.
In the sentence before, delete 'in'

We thank the reviewer. We have corrected this accordingly.

P7 'promoters more closely' - do the authors mean 'more broadly' or 'across Rap1 regulated genes'?

Indeed, the term “more closely” is not appropriate. We have substituted it (line 172):
“Next, we examined more broadly the promoters of the 141 Rap1-regulated gene for relative changes in divergent transcription (Figure 1E).”

P8 top para: greater.. (a word seems to be missing here)

We have corrected this. It reads “greater increase” (line 191).

Genome browsers shots need x axis size indicator.

The range of the x-axis is now indicated.

Fig. 2A size bar is missing

The size bar was located on top the graph. We have moved it now in the graph to make it clearer.

Fig. 5A. There is a word 'signal' in the third panel. What does this exemplify?

We thank the reviewer for pointing this out. We removed this.

Reviewer #3 (Comments to the Authors (Required)):

This paper describes an in-depth study of the mechanisms underlying divergent non-coding transcription. It is unclear how this is regulated. The authors show that the RSC complex and general regulatory factors (GRFs) contribute to repression of divergent transcription in wild type cells. Depletion of Sth1, the ATPase subunit of the RSC complex, reduces transcription of genes regulated by the Rap1 GRF; other GRFs have similar effects. The authors performed nascent RNA-seq and standard RNA-seq on wild type yeast cells and cells depleted of Sth1 or Rap1, or both. They propose that RSC-mediated nucleosome positioning over the divergent promoter is responsible for repression of divergent transcription. This is an interesting study of an important question. I have the following comments:

1. Depletion of Sth1 reduces transcription of Rap1-regulated genes by ~2-fold as measured by mRNA-seq, but there is little or no change when measured by nascent RNA-seq (Fig. 1B). This discrepancy is not discussed properly. On p.7, it is stated that, in Sth1-depleted cells, the decrease of nascent RNA is less than that of mRNA, but nascent RNA actually shows no effect (Fig. 1B left panel). Wild type control data are missing from Fig. 1B. This is the main concern - why are these two data sets giving opposite results (see #2 and #5 below also)?

We thank the reviewer for the comments. The WT-normalized data is available in figure S1C. Also, we have included that data showing the DMSO over WT in Figure S1E. As you can see there is little differences between DMSO and WT.

The reason for the differences with Sth1 depletion the nascent RNA and mRNA-seq relates to how the samples were normalized. For the Nascent RNA-seq we normalized internally, on all genes, which is the typical way of normalizing. For the mRNA-seq, we added *S. pombe* RNA as spike-in controls, to normalise and remove the effect that RSC depletion can have on bulk transcription levels at many genes (Klein-Brill et al., 2019). Noteworthy, normalization on *S. pombe* nascent RNA would have been desirable but is not straightforward to implement. We show that there still some increase in divergent transcription in the mRNA-seq (Figure 1E and S2B).

We have improved the section of the manuscript where we describe the different approaches, and added the following (lines 149 and 171):

“For mRNA-seq we added a *spike-in* control of *S. pombe* cells in a defined ratio and used it to normalize the signal of *S. cerevisiae* (Figure 1A, bottom panel). This was added to compensate for the effect that RSC depletion can have on global transcription by RNA Pol II (Parnell et al., 2008). These alterations can be normalised using external *spike-in* controls (Klein-Brill et al., 2019).. [...] We observed no decrease in the nascent RNA-seq for Rap1 regulated genes, likely because internal normalization was used for the analysis. Because of this, the nascent-RNA-seq is to be taken as a relative measurement of transcription levels throughout this manuscript. “

2. Overall, Sth1 depletion has a mild effect on anti-sense strand transcription, as measured by nascent RNA-seq (Fig. 1E, panel 1), but the effect as measured by mRNA-seq seems marginal (Fig. 1E, panel 4). Again, there is a discrepancy between the two measurements. Since this Rap1-independent effect of RSC is a key conclusion of the paper, the problem needs to be resolved. Wild-type control data should be added to Fig. 1E.

We thank the reviewer for the comment, which we addressed also in the previous point. We have included the DMSO over WT control in Figure S1E in the revised manuscript. We observed little

differences between DMSO and WT, indicating that the AID tag does not negatively impact Sth1 or Rap1 function.

While mRNA-seq measures steady state RNA, nascent RNA measures transcription directly allowing us to better measure the abundance of antisense transcripts that are known to be highly unstable. An additional factor is to the normalization strategies used, as discussed in point 1. Because of this, the nascent-RNA-seq is to be taken as a relative measurement of transcription levels throughout this manuscript.

3. What is the explanation for the increase in anti-sense transcription over the Rap1-regulated gene ORFs after Rap1 depletion (Fig. 1E, panel 5, righthand side) and, even more so, after depletion of Rap1 and Sth1 (Fig. 1E, panel 6, righthand side)?

We thank the reviewer for the comment. Depletion of Rap1 is likely leading to an increase in antisense transcription from within or 3'end of the Rap1-regulated ORFs (in addition to divergent transcription). However, this remains to be further investigated. The effect on antisense transcription is enhanced when Rap1 and Sth1 are co-depleted. It is known that RSC represses antisense transcription (Alcid & Tsukiyama 2014, Cucinotta et al., 2021). So, likely, in the absence of Rap1, RSC plays a role in repressing antisense transcription at Rap1 regulated genes.

4. The density plots in Figure 2 show peaks corresponding to about half of the number of genes plotted, which seems problematic given the area under the curve. Have the data been smoothed too much? They should really be presented as a histogram. The effect of Sth1 depletion (Fig. 2E) is small - is it just an artefact of over-smoothing? This matters because these data underpin the conclusion that RSC represses divergent transcription.

We thank the reviewer for the suggestion. We have generated the histograms of the analyses as suggested for Figure 2E (binwidth=0.3) and 3C (binwidth=0.1, included here below). We think the smoothing does not affect the conclusions as we observe the same trend. It is worth noting that the directionality score is always presented in log10 scale. For clarity reasons we prefer to keep the smoothed graph in revised manuscript.

Figure 2E

Figure 3C

Figure 3C

5. Fig. 2C shows a plot comparing fold-changes in nascent RNA-seq for the upstream RNA and the sense RNA. The effect is quite clear, but is it confirmed by the mRNA-seq data?

We thank the reviewer for the comment. We now included the same analysis for the mRNA-seq in Figure S2B. The data show similar trends as the nascent-RNA-seq. The mRNA-seq was normalized over *S. pombe* RNA, we do see a reduction in the coding direction in the mRNA-seq upon Sth1 depletion.

6. On page 12, the effect of Sth1 depletion on promoter chromatin structure is addressed (Fig. 4B,C and Fig. S4). The fragile nucleosome/MNase-sensitive complex peak observed at low MNase digestion in the PDE2 promoter (Fig. 4B, right panel) seems to be a rare exception, since there is no such peak in the average plot (Fig. 4C, righthand panel). This fact should be pointed out. Also, it is stated that "Third, the +1, -1 and -2 nucleosomes were more sensitive to MNase concentration in Q5 promoters than Q1 promoters (Figure 4C, right panel), suggesting a prevalence for partially unwrapped nucleosomes in this group." This conclusion is dubious. Since linkers are digested at different rates depending on their sequence, the relative nucleosome peak intensity at low levels of MNase may not be quantitatively reliable. Is this subtle effect reproducible? Analysis of the replicate experiments is needed. A lower nucleosome peak does not necessarily indicate fewer nucleosomes; weaker phasing flattens the peaks and fills the troughs (the extreme case is random positioning, which predicts a flat line at the average value). An alternative interpretation might be that the nucleosome-depleted region (NDR) at these promoters is widened in some Sth1-depleted cells, as the divergent promoter is activated for transcription. The assumption that fragile nucleosomes are involved also appears in the

Discussion (p. 18), without the appropriate caveat (no definitive evidence is presented here for fragile nucleosomes on divergent promoters).

We thank the reviewer for the comments. The MNase-seq data from (Kubik et al., 2018) contained sequencing data from a single data file. Because of this, we were not able to analyse the biological repeats separately. We have analysed another MNase-seq study on RSC depletion (Klein-Brill et al., 2019), now presented in Figure S4H. This dataset contains two separate replicates and shows similar nucleosome patterns as we observed in (Kubik et al., 2018). In short, the directional promoters have more defined nucleosomes at the -1/FN and -2 positions, and this positioning is affected upon Sth1 depletion. We have now quantified the presence of fragile nucleosomes as defined by (Kubik et al., 2018), and found that promoters with higher directionality have more fragile nucleosomes (Figure S4G). The fragile nucleosome positions were defined by the level of sensitivity to MNase concentrations and described in (Kubik et al., 2018).

We extended the section on interpretation of the analysis as suggested by the reviewer. We added the following paragraphs to the discussion (line 543):

“Though the effect was modest, depletion of RSC resulted in less defined consensus nucleosome patterns in the region of promoters where divergent transcription initiates. One interpretation is that lower nucleosome peaks might indicate fewer nucleosomes, which could allow the transcription machinery access to the DNA and thus result in increased divergent transcription. A more likely explanation is, that there could be weaker nucleosome phasing in RSC depleted cells, which flattens the nucleosome peaks and allows access of the transcription machinery to core promoters when the nucleosomes do not occlude the divergent promoter. Another possibility is that the NDR at directional promoters could be widened after RSC depletion, as the divergent promoter is activated and made available for transcription. Interestingly, directional promoters were also enriched for fragile nucleosomes (Figure S4G) {Brahma, 2019 #213;Kubik, 2018 #182}. Perhaps fragile nucleosomes also play a role in repressing divergent transcription, possibly by facilitating nucleosome phasing. Alternatively, it is possible that RSC indirectly represses divergent transcription by stimulating coding sense transcription. Indeed, RSC also acts in coding regions to promote Pol II elongation (Biernat et al 2021). Why RSC does not act at noncoding divergent regions cannot be explained by this interpretation. We favour a model where nucleosomes positioned by RSC form barriers to limit PIC formation and Pol II recruitment for the divergent direction (Figure 6J). Upon RSC depletion, the positioning of the -1 nucleosomes less strong and PIC formation can occur. More work is needed to disentangle how RSC and nucleosome positioning controls promoter directionality.”

7. Fig. 4D: Sth1 depletion reduces TBP levels at the promoters of genes in quintile 5. Is this small effect reproducible? Analysis of the replicate experiments is needed.

The data from the TBP ChIP-seq data were taken from Kubik et al., 2018 (Molecular Cell). This also concerned a single data file. Given that we found no other TBP ChIP-seq data available, we have decided to remove these findings from paper.

8. Discussion, page 18. It is stated that "Here, we showed that RSC mediated nucleosome positioning represses divergent noncoding transcription." What is shown is only that there is a correlation between RSC-mediated nucleosome positioning and repression of divergent noncoding transcription; cause and effect have not been distinguished.

We thank the reviewer for the comment. We agree and have changed the paragraph accordingly (line 526):

“Here, we showed that RSC represses divergent non-coding transcription. Our data are consistent with a model where RSC-mediated nucleosome positioning interferes with divergent transcription.”

9. A more general issue is that the cells are dying as essential proteins (Sth1 and/or Rap1) are depleted, perhaps activating promoters that are never activated in wild type cells. There is evidence that RSC depletion results in a general decrease in transcription (Parnell et al. 2008, cited). How might this affect the measurement of divergent transcription? The authors could comment on this in the Discussion.

We thank the reviewer for this comment. The auxin-inducible degron system rapidly and selectively depletes essential proteins, which minimises the secondary effects of the depletion on cell viability during the experiments. In our view, this is the best possible strategy to determine the direct effects of Sth1 and Rap1 depletion. There is some evidence of senescence and promoter reactivation in yeast, however, this remains not well understood and seems a rather slow process requiring growth for many generations (<https://www.ncbi.nlm.nih.gov/pmc/articles/PMC3701195/>). We have now included a section in the discussion describing this issue of transcription levels and RSC depletion (line 606):

“The nascent RNA-seq data in this manuscript was normalized on all genes, and thus should be considered a relative measurement. Since RSC controls global transcription by Pol II, the relative nature of the nascent RNA-seq measurement is in particular relevant for the RSC depletion analysis (Parnell et al 2008). For this reason, we compared the divergent transcription relative to coding gene transcription throughout this manuscript. Noteworthy, the mRNA-seq analysis normalized on external spike-ins, showed, albeit less, also increased divergent non-coding RNA expression upon RSC depletion. Irrespective of whether the effects of RSC depletion are absolute or relative, evidently divergent non-coding transcription is controlled by RSC in a distinct way compared to coding gene transcription.”

Minor points:

10. I found the term "divergent transcription" confusing, because divergent transcription is often used to mean transcription of two genes from a central promoter, implying that transcription occurs in both directions. Here, divergent transcription refers to transcription of the gene upstream of the Rap1-regulated gene (i.e. transcription of the upstream gene only). For example, in the section title "RSC represses divergent transcription independently of Rap1", the authors are presumably referring only to the upstream gene, not both genes (since transcription of the downstream, Rap1-regulated, gene is activated by RSC, not repressed; Fig. 1B). However, I don't know how to fix the terminology!

We agree that RSC-dependent divergent noncoding transcripts are upstream of the Rap1 binding sites. Indeed, the divergent transcription occurs formally upstream of the Rap1-regulated promoter sequences. Often, however, it is not clearly defined where a promoter sequence exactly starts or stops.

The term “divergent transcription” refers to transcription upstream of the main TSS in the antisense direction, while “coding transcription” refers to transcription in the sense direction. Reviewer 1 also made a similar valid point regarding the relative nature of the nascent-RNA-seq. Hence, we have now stated where needed that changes are relative to the coding gene transcription.

11. The IAA concentration used is missing.

We have now added this to main text (line 145).

12. Fig. 1A: some bands seem to have been cut off in the Western.

We have fixed the organization of the western blot in Figure 1A. We have also included the uncropped data of all the western blots and northern blot as supplemental data.

13. The double-depletion data (Sth1 and Rap1) are missing from Fig. 1D.

We thank the reviewer for pointing this out. We have now correctly included the double depletion data for Figure 1D.

References:

- Alcid EA, Tsukiyama T. 2014. Atp-dependent chromatin remodeling shapes the long noncoding rna landscape. *Genes Dev.* 28(21):2348-2360. doi:10.1101/gad.250902.114
- Biernat E, Kinney J, Dunlap K, Rizza C, Govind CK. 2021. The rsc complex remodels nucleosomes in transcribed coding sequences and promotes transcription in *saccharomyces cerevisiae*. *Genetics.* 217(4) doi:10.1093/genetics/iyab021
- Brahma S, Henikoff S. 2019. Rsc-associated subnucleosomes define mnase-sensitive promoters in yeast. *Mol Cell.* 73(2):238-249 e233. doi:10.1016/j.molcel.2018.10.046
- Challal D, Barucco M, Kubik S, Feuerbach F, Candelli T, Geoffroy H, Benaksas C, Shore D, Libri D. 2018. General regulatory factors control the fidelity of transcription by restricting non-coding and ectopic initiation. *Mol Cell.* 72(6):955-969 e957. doi:10.1016/j.molcel.2018.11.037
- Cucinotta CE, Dell RH, Bracerros KC, Tsukiyama T. 2021. Rsc primes the quiescent genome for hypertranscription upon cell-cycle re-entry. *Elife.* 10 doi:10.7554/eLife.67033
- Jin Y, Eser U, Struhl K, Churchman LS. 2017. The ground state and evolution of promoter region directionality. *Cell.* 170(5):889-898 e810. doi:10.1016/j.cell.2017.07.006
- Klein-Brill A, Joseph-Strauss D, Appleboim A, Friedman N. 2019. Dynamics of chromatin and transcription during transient depletion of the rsc chromatin remodeling complex. *Cell Rep.* 26(1):279-292 e275. doi:10.1016/j.celrep.2018.12.020
- Kubik S, Bruzzone MJ, Challal D, Dreos R, Mattarocci S, Bucher P, Libri D, Shore D. 2019. Opposing chromatin remodelers control transcription initiation frequency and start site selection. *Nat Struct Mol Biol.* 26(8):744-754. doi:10.1038/s41594-019-0273-3
- Kubik S, O'Duibhir E, de Jonge WJ, Mattarocci S, Albert B, Falcone JL, Bruzzone MJ, Holstege FCP, Shore D. 2018. Sequence-directed action of rsc remodeler and general regulatory factors modulates +1 nucleosome position to facilitate transcription. *Mol Cell.* 71(1):89-102 e105. doi:10.1016/j.molcel.2018.05.030
- Wu ACK, Patel H, Chia M, Moretto F, Frith D, Snijders AP, van Werven FJ. 2018. Repression of divergent noncoding transcription by a sequence-specific transcription factor. *Mol Cell.* 72(6):942-954 e947. doi:10.1016/j.molcel.2018.10.018
- Biernat E, Kinney J, Dunlap K, Rizza C, Govind CK. 2021. The rsc complex remodels nucleosomes in transcribed coding sequences and promotes transcription in *saccharomyces cerevisiae*. *Genetics.* 217(4) doi:10.1093/genetics/iyab021
- Parnell TJ, Huff JT, Cairns BR. 2008. Rsc regulates nucleosome positioning at pol ii genes and density at pol iii genes. *EMBO J.* 27(1):100-110. doi:10.1038/sj.emboj.7601946

August 26, 2022

RE: Life Science Alliance Manuscript #LSA-2022-01394R

Dr. Folkert van Werven
The Francis Crick Institute
1 Midland Road
London NW1 1AT
United Kingdom

Dear Dr. van Werven,

Thank you for submitting your revised manuscript entitled "RSC and GRFs confer promoter directionality by restricting divergent noncoding transcription". We would be happy to publish your paper in Life Science Alliance pending final revisions necessary to meet our formatting guidelines.

- please address Reviewer 1's remaining points
- given the Reviewers' input, it does seem best to avoid commenting on fragile nucleosomes
- please add a Summary Blurb/Alternate Abstract in our system
- please upload your main and supplementary figures as single files
- please add a callout for Each Figure to your main manuscript text (Figure 2 (D,E), Figure S2A, Figure 3A, Figure 4D, Figure S4 (D,E,F), Figure S5C, Figure 6A)
- please add the Twitter handle of your host institute/organization as well as your own or/and one of the authors in our system
- please add sizes next to all blots
- please list figure legends in the order of: all main figure legends, followed by all supplemental figure legends
- please upload one source data file per relevant figure, labeled accordingly
- please upload the Supplemental Tables as individual files, labeled accordingly

A. FINAL FILES:

B. MANUSCRIPT ORGANIZATION AND FORMATTING:

Sincerely,

Reviewer #1 (Comments to the Authors (Required)):

The revised manuscript is improved in many ways. I did not have time to examine methods and figure legends of the revision, but the authors did make good faith attempts at addressing reviewer concerns.

Comments on the revision:

1. Discussion is relatively long and could be more concise/focused.
2. line 214. "Figure S3A"- there is only Figure S3. Please check all figure callouts carefully.
3. line 295 Figure 5B- enrichment of A tracts: 5B suggests A tracts are different places not necessarily differentially enriched-unless it is meant that there is one peak vs two?
4. line 317. "(Figure 5B)"- is this meant to be 5D?
5. line 361 "We conclude that, as we reported previously for Rap1 (Wuet al 2018), other transcription factors can repress divergent transcription when targeted to regions close to their binding sites."

The above sentence seems non-sensical. "other...factors can repress...when targeted to regions close to their binding sites". I think what is meant is that TFs can repress transcription if appropriately positioned adjacent to TSSs?

6. line 427 "Upon RSC depletion, the positioning of the -1 nucleosomes [is?] less strong and PIC formation can occur." Is there a missing word?

Cross comments:

There are data from Henikoff lab that argue that the MNase sensitive features of promoters are in fact nucleosomes, and this is done with a CUT&RUN+pull down approach. It would be appropriate to be careful about how to approach different data sets

Reviewer #3 (Comments to the Authors (Required)):

The revised manuscript is much improved.

I have only one comment and that concerns my original point #6. The authors were dependent on the data from Kubik et al. (cited) which claimed the presence of so-called fragile nucleosomes at yeast promoters. Unfortunately, as the authors point out, Kubik et al. did not do biological replicate experiments(!) and, in fact, their data are of poor quality. In the revised manuscript, the authors use data from Klein-Brill et al. (cited), but Klein-Brill et al. did not do the MNase digestion series necessary to detect MNase-sensitive complexes (actually, they don't mention fragile nucleosomes at all). Consequently, I don't see how the authors can discuss fragile nucleosomes using these data. I suggest that the authors avoid commenting on fragile nucleosomes altogether. Alternatively, the authors could analyse data from Chereji et al. (cited); this paper confirms the presence of MNase-sensitive complexes at promoters, but shows that they are not nucleosomes of any kind because they don't contain histones (biological replicates are provided). The paper by Brahma et al. (cited) shows that RSC-nucleosome complexes can be detected at promoters but at much lower frequency than MNase-sensitive complexes.

-please address Reviewer 1's remaining points

Done, see below.

-given the Reviewers' input, it does seem best to avoid commenting on fragile nucleosomes

We have removed the term from the manuscript.

-please add a Summary Blurb/Alternate Abstract in our system

Done.

-please upload your main and supplementary figures as single files

Done.

-please add a callout for Each Figure to your main manuscript text (Figure 2 (D,E), Figure S2A, Figure 3A, Figure 4D, Figure S4 (D,E,F), Figure S5C, Figure 6A)

Done.

-please add the Twitter handle of your host institute/organization as well as your own or/and one of the authors in our system

-please add sizes next to all blots

Done.

-please list figure legends in the order of: all main figure legends, followed by all supplemental figure legends

Done.

-please upload one source data file per relevant figure, labeled accordingly

Done.

-please upload the Supplemental Tables as individual files, labeled accordingly

Done.

Reviewer #1 (Comments to the Authors (Required)):

The revised manuscript is improved in many ways. I did not have time to examine methods and figure legends of the revision, but the authors did make good faith attempts at addressing reviewer concerns.

Comments on the revision:

1. Discussion is relatively long and could be more concise/focused.

We have shortened and reorganized the discussion.

2. line 214. "Figure S3A"- there is only Figure S3. Please check all figure callouts carefully.

We have corrected this.

3. line 295 Figure 5B- enrichment of A tracts: 5B suggests A tracts are different places not necessarily differentially enriched- unless it is meant that there is one peak vs two?

The reference is to the table s1. We now corrected this.

"We found that A-tracks and the CG(C/G)G motif were enriched in the group of promoters with the highest directionality (Q5) compared to the group of promoters with lowest directionality (Q1) (Table S1)."

4. line 317. "(Figure 5B)"- is this meant to be 5D?

5C. we have corrected this.

5. line 361 "We conclude that, as we reported previously for Rap1 (Wuet al 2018), other transcription factors can repress divergent transcription when targeted to regions close to their binding sites."

The above sentence seems non-sensical. "other...factors can repress...when targeted to regions close to their binding sites". I think what is meant is that TFs can repress transcription if appropriately positioned adjacent to TSSs?

We have now rephrased this.

We conclude that, as we reported previously for Rap1 (Wu et al 2018), transcription factors can repress divergent transcription if appropriately positioned adjacent to divergent TSSs.

6. line 427 "Upon RSC depletion, the positioning of the -1 nucleosomes [is?] less strong and PIC formation can occur." Is there a missing word?

Corrected.

Cross comments:

There are data from Henikoff lab that argue that the MNase sensitive features of promoters are in fact nucleosomes, and this is done with a CUT&RUN+pull down approach. It would be appropriate to be careful about how to approach different data sets

We agree with the Henikoff lab observation. However, since reviewer 3 disagrees and the field is still split on the topic of fragile nucleosomes. We have removed the fragile nucleosome quantification and avoided the term through the manuscript.

Reviewer #3 (Comments to the Authors (Required)):

The revised manuscript is much improved.

I have only one comment and that concerns my original point #6. The authors were dependent on the data from Kubik et al. (cited) which claimed the presence of so-called fragile nucleosomes at yeast promoters. Unfortunately, as the authors point out, Kubik et al. did not do biological replicate experiments(!) and, in fact, their data are of poor quality. In the revised manuscript, the authors use data from Klein-Brill et al. (cited), but Klein-Brill et al. did not do the MNase digestion series necessary to detect MNase-sensitive complexes (actually, they don't mention fragile nucleosomes at all). Consequently, I don't see how the authors can discuss fragile nucleosomes using these data. I suggest that the authors avoid commenting on fragile nucleosomes altogether. Alternatively, the authors could analyse data from Chereji et al. (cited); this paper confirms the presence of MNase-

sensitive complexes at promoters, but shows that they are not nucleosomes of any kind because they don't contain histones (biological replicates are provided). The paper by Brahma et al. (cited) shows that RSC-nucleosome complexes can be detected at promoters but at much lower frequency than MNase-sensitive complexes.

We agree with the Henikoff lab observation. However, since reviewer 3 disagrees and the field is still split on the topic of fragile nucleosomes. We have removed the fragile nucleosome quantification and avoided the term through the manuscript.

September 2, 2022

RE: Life Science Alliance Manuscript #LSA-2022-01394RR

Dr. Folkert van Werven
The Francis Crick Institute
1 Midland Road
London NW1 1AT
United Kingdom

Dear Dr. van Werven,

Thank you for submitting your Research Article entitled "RSC and GRFs confer promoter directionality by restricting divergent noncoding transcription". It is a pleasure to let you know that your manuscript is now accepted for publication in Life Science Alliance. Congratulations on this interesting work.

DISTRIBUTION OF MATERIALS:

Again, congratulations on a very nice paper. I hope you found the review process to be constructive and are pleased with how the manuscript was handled editorially. We look forward to future exciting submissions from your lab.

Sincerely,
